# Highly Efficient and Effective LLMs with Multi-Boolean Architectures

**Ba-Hien Tran & Van Minh Nguyen**
Huawei Paris Research Center
Paris, France
`ba.hien.tran@huawei.com` *(corresponding author)*

## Abstract

Weight binarization has emerged as a promising strategy to reduce the complexity of large language models (LLMs). Existing approaches fall into post-training binarization, which is simple but causes severe performance loss, and training-aware methods, which depend on full-precision latent weights, adding complexity and limiting efficiency. We propose a novel framework that represents LLMs with multi-kernel Boolean parameters and, for the first time, enables direct finetuning LMMs in the Boolean domain, eliminating the need for latent weights. This enhances representational capacity and dramatically reduces complexity during both finetuning and inference. Extensive experiments across diverse LLMs show our method outperforms recent ultra low-bit quantization and binarization techniques.

## 1 Introduction

Large language models (Brown et al., 2020; Touvron et al., 2023a; Liu et al., 2024a) have demonstrated unprecedented capabilities, largely due to the continuous growth in both model and dataset sizes. A key area of focus in optimizing these models is lower-precision computation, which offers substantial benefits in terms of memory and computational efficiency. One prominent approach to achieving this is through the quantization of weight parameters, which reduces the model size by lowering the precision of the weight values. Recent studies on scaling laws (Dettmers & Zettlemoyer, 2023; Kumar et al., 2025) have highlighted the potential of using low-precision techniques for large language models (LLMs).

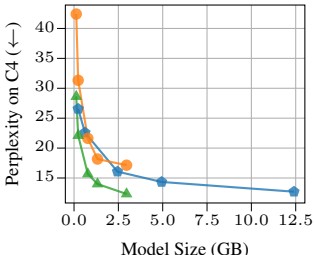

Figure 1: Finetuning OPT models (Zhang et al., 2022) using our 3 Boolean kernels ( ▲ ), compared to OPTQ (Frantar et al., 2023) ( ▲ ), which quantizes the models to 3 bits, and the FP16 baseline ( ⬟ ) on the C4 dataset.

Binarization represents one of the most extreme forms of quantization for LLMs. While significant progress has been made, challenges remain (Yuan et al., 2024; Huang et al., 2024; Li et al., 2025). Even with advanced techniques like Quantization-Aware Training (QAT), which fine-tunes the model extensively after binarization (Xu et al., 2024; Jo et al., 2024), or trains it from scratch (Wang et al., 2023), performance still lags behind that of full-precision (FP) models. This performance gap can be attributed to the limited representation capacity of binary weights and the heavy reliance on FP latent weights for binarization. This reliance not only makes the approach computationally expensive but also suboptimal, as it requires gradient approximation. Meanwhile, recent advances in 4-bit quantization have achieved remarkable compression with minimal accuracy loss, but further compression or applying these methods to smaller models has yielded unsatisfactory results (Frantar et al., 2023; Lin et al., 2024).

In this paper, we aim to push the boundary of low-precision LLMs by proposing a novel method named as Multiple Boolean Kernels (MBOK). We extend the work in Nguyen et al. (2024), which proposes training neural networks with native Boolean weights directly in the Boolean domain, However, effectively applying this approach to LLMs remains a key challenge. In particular, our contributions are:

- We propose the framework MBOK, which employs multiple Boolean kernels, each using distinct Boolean weights (§ 4.2). This allows for flexibly representing LLMs with low bits, while approaching to FP performance with minimal *both* finetuning and inference cost. The Boolean weights are directly trained in Boolean domain, avoiding the need for FP latent weights and gradient approximations.

- We propose a novel successive method that effectively transfers knowledge from an FP LLM into the Boolean model (§ 4.3), followed by further fine-tuning using knowledge distillation (§ 4.3.2).
- We introduce a method for automatically allocating the number of kernels for each weight (§ 5), supporting any average bit-width, including fractional values.
- We provide a comprehensive empirical analysis and benchmarks, demonstrating our method's superior performance over recent binarization and quantization approaches (see § 6) with much lower memory and computational overhead. For example, Fig. 1 shows that our method achieves the best accuracy-compression trade-off, outperforming FP and existing quantization techniques.

## 2 RELATED WORKS

**LLMs quantization.** Quantization techniques are commonly used to reduce the memory and latency of LLMs. They fall into two categories: QAT, which involves retraining or finetuning in a quantized form, and Post-Training Quantization (PTQ), which can be applied directly without retraining. Due to the difficulty of retraining such large models, most work focuses on PTQ (Frantar et al., 2023; Sheng et al., 2023; Lin et al., 2024; Lee et al., 2024), though recent efforts also explore QAT via data-free methods (LLM-QAT (Liu et al., 2024c)), or parameter-efficient fine-tuning like LoRA (Dettmers et al., 2023). A promient PTQ method is OPTQ (Frantar et al., 2023), which introduces one-shot low-bit weight quantization using approximate second-order information. Follow-up work refines this by addressing outliers (Kim et al., 2024; Dettmers et al., 2024), accounting for activation effects (Lin et al., 2024; Lee et al., 2024), and optimizing quantization parameters (OmniQuant (Shao et al., 2024)). However, effective LLMs quantization is still challenging (Xu et al., 2025).

**Binarization.** This represents the most extreme form of quantization, typically using the $\mathbf{sign}(\cdot)$ function with gradients estimated via the straight-through-estimator (STE) (Bengio et al., 2013). Early work focused on small Transformer models (Vaswani et al., 2017) trained or fine-tuned on labeled data (Bai et al., 2021; Qin et al., 2022; Liu et al., 2022; 2023). Recent efforts have extended binarization to LLMs. Methods like BiLLM (Huang et al., 2024), PB-LLM (Yuan et al., 2024), STBLLM (Dong et al., 2025), and ARB-LLM (Li et al., 2025) adopt hybrid PTQ approaches, binarizing non-salient weights while using higher precision for important ones, with calibration data used to adjust scaling factors. BitStack (Wang et al., 2025), QBB (Bulat et al., 2024), DB-LLM (Chen et al., 2024) further improve this with multiple binary bases, either through a training-free method or via knowledge distillation. In contrast, BitNet (Wang et al., 2023) replaces linear layers with a custom 1-bit weight structure, BitLinear, and trains the model from scratch. OneBit (Xu et al., 2024), which decomposes weights into 1-bit components and scaling vectors for QAT, further enhanced by MoS (Jo et al., 2024) using a mixture of scalings. Despite progress, these methods remain costly due to their dependence on FP latent weights during training. Table 1 summarizes the key characteristics of these methods in comparison to ours.

Table 1: A summary of SOTA binarization methods for LLMs compared to our method.

| Method | Train from Scratch | Post-training Binarization | Finetune from FP Model | Calibration Data | Weight Update | Multiple Binary Bases | Higher-bit Salient Weights |
|---|---|---|---|---|---|---|---|
| BitNet (Wang et al., 2023) | ✓ | ✗ | ✗ | NA | FP latent-weights | ✗ | ✗ |
| BiLLM (Huang et al., 2024) | ✗ | ✓ | ✗ | ✓ | NA | ✓ | ✓ |
| PB-LLM (Yuan et al., 2024) | ✗ | ✓ | ✗ | ✓ | NA | ✗ | ✓ |
| STBLLM (Dong et al., 2025) | ✗ | ✓ | ✗ | ✓ | NA | ✓ | ✓ |
| ARB-LLM (Li et al., 2025) | ✗ | ✓ | ✗ | ✓ | NA | ✓ | ✓ |
| BitStack (Wang et al., 2025) | ✗ | ✓ | ✗ | ✗ | NA | ✓ | ✗ |
| DB-LLM (Chen et al., 2024) | ✗ | ✓ | ✓ | ✓ | FP latent-weights | ✓ | ✗ |
| QBB (Bulat et al., 2024) | ✗ | ✓ | ✓ | ✓ | FP latent-weights | ✓ | ✗ |
| OneBit (Xu et al., 2024) | ✗ | ✗ | ✓ | ✓ | FP latent-weights | ✗ | ✗ |
| MoS (Jo et al., 2024) | ✗ | ✗ | ✓ | ✓ | FP latent-weights | ✗ | ✗ |
| MBOK [Ours] | ✗ | ✗ | ✓ | ✓ | Native Boolean weights | ✓ | ✗ |

## 3 PRELIMINARIES

**Notations.** We use a standard notation for vectors ($\mathbf{a}$), matrices ($\mathbf{A}$), and scalars ($a$). The $i$-th element of a vector $\mathbf{a}$ is $\mathbf{a}_{[i]}$, and the element at the $i$-th row and $j$-th column of a matrix $\mathbf{A}$ is $\mathbf{A}_{[i,j]}$. The symbol $\odot$ denotes element-wise multiplication, with broadcasting if needed.

### 3.1 PITFALLS OF FULL-PRECISION LATENT WEIGHTS FOR BINARIZATION

Binarization is an effective technique for reducing both the size and computation of deep learning models by converting high-precision weight parameters into 1-bit values (Hubara et al., 2016;

Courbariaux et al., 2015; Rastegari et al., 2016). For a linear layer, $\mathbf{Y} = \mathbf{X}\mathbf{W}_{\text{FP}}^{\top} + \mathbf{b}$, where $\mathbf{X}_{\text{FP}} \in \mathbb{R}^{b \times n}$ is the input data, and $\mathbf{W} \in \mathbb{R}^{m \times n}$ with the input size $n$ and output size $m$, and $\mathbf{b} \in \mathbb{R}^m$ are the FP weights and bias. Binarization results in $\mathbf{Y} = \alpha \cdot \mathbf{X}\mathbf{W}_{\text{bin}}^{\top} + \mathbf{b}$, with $\mathbf{W}_{\text{bin}} = \mathbf{sign}(\mathbf{W}_{\text{FP}})$ and $\alpha$ as a scaling factor (e.g., $\alpha = \frac{\|\mathbf{W}_{\text{FP}}\|_1}{m \times n}$) (Rastegari et al., 2016).

During training, the FP weights must be retained for learning the binarized weights. In vanilla gradient descent, binarized weights are updated as $\mathbf{W}_{\text{bin}} = \mathbf{sign}(\mathbf{W}_{\text{FP}} - \eta \cdot \mathbf{G}_{\mathbf{W}_{\text{FP}}})$, where $\eta$ is the learning rate and $\mathbf{G}_{\mathbf{W}_{\text{FP}}}$ is the gradient of the FP weights. This leads to high memory usage, especially with optimizers like Adam (Kingma & Ba, 2015), which require storing two additional FP momenta for each parameter. Moreover, the gradient approximation for binarized weights often uses a differentiable proxy, like the STE (Bengio et al., 2013), but this introduces performance drops due to proxy gradient noise. This noise can cause oscillations and instability during training.

### 3.2 NATIVE BOOLEAN FRAMEWORK FOR NEURAL NETWORKS

To address the issues associated with latent-weight-based approaches, Nguyen et al. (2024) recently proposed a principled framework for directly training Boolean neural networks in the Boolean domain. Consider the $l$-th Boolean linear layer; in the forward pass, the output of the next layer is defined as:

$$\mathbf{Y}_{[k,j]}^{(l)} = \mathbf{b}_{[j]}^{(l)} + \sum_{i=1}^{n} \mathrm{L}(\mathbf{X}_{[k,i]}^{(l)}, \mathbf{W}_{[i,j]}^{(l)}), \qquad 1 \le j \le m, \tag{1}$$

where $k$ denotes the sample index in the batch, and $\mathrm{L}$ is a logic gate such as $\mathbf{and}$, $\mathbf{or}$, $\mathbf{xor}$, or $\mathbf{xnor}$; Hereafter, for clarity, we consider $\mathrm{L} = \mathbf{xnor}$ as a concrete example. The weights $\mathbf{W}_{[i,j]}^{(l)}$ are Boolean values $\{\text{TRUE}, \text{FALSE}\}$ or $\{-1, +1\}$, as typically used in practical implementations.

The logic gate $\mathrm{L}$ can be extended to handle mixed-type data. In this paper, we focus on the case where the input data is real-valued, and the weights are Boolean. Specifically, for an input element $x \in \mathbb{R}$, we define $x_{\text{bool}} = \text{TRUE} \Leftrightarrow x \ge 0$, and $x_{\text{bool}} = \text{FALSE} \Leftrightarrow x < 0$, and $|x|$ its magnitude. The logic operation between a real input $x \in \mathbb{R}$ and a Boolean weight $w \in \mathbb{B}$ is defined as $\mathbf{xnor}(w, x) \triangleq s$ such that $s_{\text{bool}} = \mathbf{xnor}(w_{\text{bool}}, x)$ and $|s| = |x|$.

**Backward pass.** This layer receives the backpropagation signal from the downstream layer. Specifically, $\mathbf{Z}_{[k,j]}^{(l)} \triangleq \frac{\delta \mathcal{L}}{\delta \mathbf{Y}_{[k,j]}^{(l)}}$ denotes the variation of the loss function $\mathcal{L}$ w.r.t. the output at layer $l$. To optimize the Boolean weights, we need to compute the corresponding loss signal, denoted as $\mathbf{Q}_{[i,j]}^{(l)} \triangleq \frac{\delta \mathcal{L}}{\delta \mathbf{W}_{[i,j]}^{(l)}}$, which is aggregated over the batch dimension $k$ as:

$$\mathbf{Q}_{[i,j]}^{(l)} = \sum_{k=1}^{b} \mathbf{1}(\mathbf{Q}_{[k,i,j]}^{(l)} = \text{TRUE})|\mathbf{Q}_{[k,i,j]}^{(l)}| - \sum_{k=1}^{b} \mathbf{1}(\mathbf{Q}_{[k,i,j]}^{(l)} = \text{FALSE})|\mathbf{Q}_{[k,i,j]}^{(l)}|, \tag{2}$$

where $\mathbf{Q}_{[i,j,k]}^{(l)} = \mathbf{xnor}(\mathbf{Z}_{[k,j]}^{(l)}, \mathbf{X}_{[k,i]}^{(l)})$, and $\mathbf{1}(\cdot)$ is the indicator function. The backpropagation signal for the upstream layer, $\mathbf{P}_{[k,j]}^{(l)} \triangleq \frac{\delta \mathcal{L}}{\delta \mathbf{X}_{[k,j]}^{(l)}}$, can be computed in a similar manner.

**Boolean optimizer.** Given the loss signal, the rule for updating the Boolean weight $\mathbf{W}_{[i,j]}^{(l)}$ to minimize the loss function $\mathcal{L}$ is as $\mathbf{W}_{[i,j]}^{(l)} = \neg \mathbf{W}_{[i,j]}^{(l)}$ if $\mathbf{xnor}(\mathbf{Q}_{[i,j]}^{(l)}, \mathbf{W}_{[i,j]}^{(l)}) = \text{TRUE}$. Based on this update rule, we can develop an optimizer that accumulates the signal $\mathbf{Q}_{[i,j]}^{(l)}$ over training iterations. Specifically, let $\mathbf{W}_{[i,j]}^{(l),t}$ denotes the weight at iteration $t$, and $\mathbf{M}_{[i,j]}^{(l),t}$ represents its accumulator, initialized as $\mathbf{M}_{[i,j]}^{(l),0} = 0$. The update rule for the accumulator is then defined as:

$$\mathbf{M}_{[i,j]}^{(l),t+1} \leftarrow \beta^t \mathbf{M}_{[i,j]}^{(l),t} + \eta \mathbf{Q}_{[i,j]}^{(l),t}, \tag{3}$$

where $\eta$ is the accumulation factor acting as a learning rate, and $\beta^t$ is a regularizing factor that reflects the system's state at time $t$. In our work, we use brain plasticity (Fuchs et al., 2014) and Hebbian theory (Hebb, 2005) to adaptively set $\beta^t$. We encourage the reader check Appendix A for details.

**Remarks on complexity and applicability to LLMs.** This Boolean framework optimizes Boolean parameters $\mathbf{W}_{[i,j]}^{(l)}$ directly in the Boolean space, eliminating the need for FP latent weights. As shown in Eq. 3, the Boolean optimizer is more lightweight than common LLM optimizers like Adam,

requiring only one FP momentum per parameter. This reduces both training and inference complexity and avoids gradient approximation induced from STE. As shown in Proposition A.10 in Appendix, $\mathbf{xnor}(w, s) = w \times s$, mathematically enabling direct application to existing linear algebra operations. Practically, native logic operations are much faster than multiplication.

## 4 MULTIPLE BOOLEAN KERNELS

### 4.1 BOOLEAN REFORMULATION FOR LINEAR LAYERS

LLMs (Brown et al., 2020) are mostly based on the Transformer architecture (Vaswani et al., 2017), in which linear layers are the core elements. Inpsired by Xu et al. (2024), we employ sign-value-independent decomposition (SVID) such that an FP input matrix $\mathbf{W} \in \mathbb{R}^{m \times n}$ of linear layers is decomposed into one Boolean matrix $\mathbf{W}_{\mathrm{bool}} \triangleq \mathbf{sign}(\mathbf{W})$ and two FP

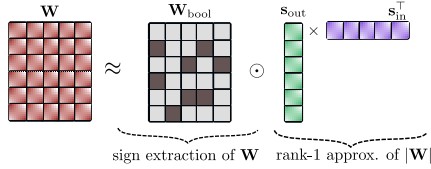

Figure 2: Illustration of SVID.

vectors $\mathbf{s}_{\mathrm{in}}$ and $\mathbf{s}_{\mathrm{out}}$. Precisely, let $|\mathbf{W}|$ be the element-wise absolute value of $\mathbf{W}$, write $|\mathbf{W}| = \mathbf{U}\mathbf{\Sigma}\mathbf{V}^{\top}$ its singular value decomposition (SVD) (Beltrami, 1990). Using rank-1 approximation of $|\mathbf{W}|$, $\mathbf{s}_{\mathrm{in}}$ and $\mathbf{s}_{\mathrm{out}}$ are given as: $\mathbf{s}_{\mathrm{in}} = \sqrt{\sigma_1}\mathbf{V}_{[:,1]}$, and $\mathbf{s}_{\mathrm{out}} = \sqrt{\sigma_1}\mathbf{U}_{[:,1]}$. Then, the input matrix is approximated as $\mathbf{W} = \mathbf{W}_{\mathrm{bool}} \odot |\mathbf{W}| \approx \mathbf{W}_{\mathrm{bool}} \odot (\mathbf{s}_{\mathrm{out}}\mathbf{s}_{\mathrm{in}}^{\top})$. This procedure is illustrated in Fig. 2.

**Proposition 4.1.** *(Xu et al., 2024) For* $\mathbf{W} \in \mathbb{R}^{m \times n}$*, write* $\mathbf{W} = \widetilde{\mathbf{U}}\widetilde{\mathbf{\Sigma}}\widetilde{\mathbf{V}}^{\top}$ *its SVD. Let* $\mathbf{a} = \sqrt{\widetilde{\sigma}_1}\widetilde{\mathbf{U}}_{[:,1]}$*, and* $\mathbf{b} = \sqrt{\widetilde{\sigma}_1}\widetilde{\mathbf{V}}_{[:,1]}$*. With the notations as described above, we have:*

$$\left\|\mathbf{W} - \mathbf{W}_{\mathrm{bool}} \odot \mathbf{s}_{\mathrm{out}}\mathbf{s}_{\mathrm{in}}^{\top}\right\|_F^2 \leq \left\|\mathbf{W} - \mathbf{a}\mathbf{b}^{\top}\right\|_F^2. \tag{4}$$

*Remark* 4.2. Proposition 4.1 re-states Proposition 2 of Xu et al. (2024) with its precise assumption of vectors $\mathbf{a}$ and $\mathbf{b}$ which is necessary for its proof provided in Appendix therein.

Proposition 4.1 shows that using $\mathbf{W}_{\mathrm{bool}}$ together with value matrix approximation is better than a direct rank-1 approximation of $\mathbf{W}$ in terms of Frobenius-norm. This emphasizes the important role of $\mathbf{W}_{\mathrm{bool}}$ in approximating the original FP matrix. Moreover, our following Proposition 4.3 shows that the SVID approximation as described above is optimal for approximating the original matrix $\mathbf{W}_{\mathrm{bool}}$.

**Proposition 4.3.** *For* $\mathbf{W} \in \mathbb{R}^{m \times n}$ *and the notations as described above, we have:*

$$\left\|\mathbf{W} - \mathbf{W}_{\mathrm{bool}} \odot \mathbf{s}_{\mathrm{out}}\mathbf{s}_{\mathrm{in}}^{\top}\right\|_F^2 \leq \left\|\mathbf{W} - \mathbf{W}_{\mathrm{bool}} \odot \mathbf{c}\mathbf{d}^{\top}\right\|_F^2, \quad \forall \mathbf{c} \in \mathbb{R}^{m \times 1}, \forall \mathbf{d} \in \mathbb{R}^{n \times 1}. \tag{5}$$

The proof is given in Appendix D.3. The linear layer can be then reformulated as (Xu et al., 2024):

$$\mathbf{X}\mathbf{W}_{\mathrm{FP}}^{\top} \approx \left[\left(\mathbf{X} \odot \mathbf{s}_{\mathrm{in}}^{\top}\right) \mathbf{W}_{\mathrm{bool}}\right] \odot \mathbf{s}_{\mathrm{out}}^{\top}. \tag{6}$$

### 4.2 ENHANCED EXPRESSIVITY WITH MULTIPLE BOOLEAN KERNELS

We have shown that SVID provides a good approximation of the original weights, its expressivity can be still limited to capture well the original FP parameters of complicated models, which were trained on large-scale datasets over extended periods of time. To overcome this limitation, we propose the use of a multi-Boolean kernel structure for the weights. Specifically, we employ $K$ kernels, where each kernel utilizes distinct Boolean weights and scaling factors, to better represent the original weight parameters. This leads to the approximation: $\mathbf{W}_{\mathrm{FP}} \approx \mathbf{W}_{\mathrm{approx}} \triangleq$

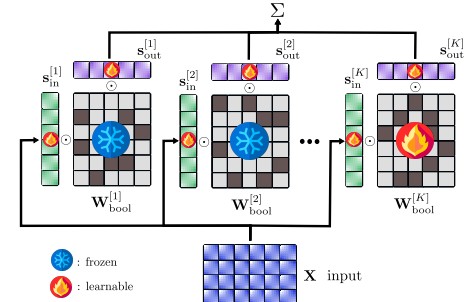

Figure 3: The computation of a linear layer approximated using multi kernels of Boolean.

$\sum_{k=1}^{K} \mathbf{W}_{\mathrm{bool}}^{[k]} \odot (\mathbf{s}_{\mathrm{out}}^{[k]}\mathbf{s}_{\mathrm{in}}^{[k]\top})$. The computation of a linear layer can then be approximated as follows (see Fig. 3 for an illustration):

$$\mathbf{X}\mathbf{W}_{\mathrm{FP}}^{\top} \approx \sum_{k=1}^{K} \left[\left(\mathbf{X} \odot \mathbf{s}_{\mathrm{in}}^{[k]\top}\right) \mathbf{W}_{\mathrm{bool}}^{[k]}\right] \odot \mathbf{s}_{\mathrm{out}}^{[k]\top}. \tag{7}$$

Here, the computational costs associated with the FP scaling factors, $\mathbf{s}_{\text{in}}$ and $\mathbf{s}_{\text{out}}$, are small because they only involve element-wise multiplications. The dominant computational cost arises from the matrix multiplication between the scaled input data, $\mathbf{X} \odot \mathbf{s}_{\text{in}}$, and the weights. However, thanks to the use of Boolean weights, the complexity is significantly reduced, as these multiplications can be replaced by additions. Moreover, as we will demonstrate in § 6.1.1, only a small number of kernels are required to achieve a reasonable result. Additionally, we find that, after the successive extraction process from the FP model (§ 4.3.1), we only need to train the Boolean weights for the last kernel and the scaling factors, further significantly reducing the overall complexity.

### 4.3 EFFECTIVE KNOWLEDGE TRANSFER INTO BOOLEAN MODELS

We have introduced our proposed multi-Boolean kernel structure for effectively representing the linear layers of LLMs. In this section, we outline the process for transferring knowledge from a source FP model to a Boolean model. This process consists of two steps: (1) data-free initialization to maximize information retention from the source, and (2) data-dependent finetuning, where the Boolean model is further trained on a target dataset with guidance from the FP model.

#### 4.3.1 SUCCESSIVE EXTRACTION USING SVID

For each linear layer, to initialize the values of the Boolean weights and scaling factors for all kernels, we successively apply SVID to the given FP weights. The goal here is to further proceed to SVID process to approximate the residual error introduced by the previous step. Specifically, after each step of decomposing the weight matrix using SVID, we obtain a residual matrix, which is defined as:

$$\mathbf{W}_{\text{res}}^{[k]} = \mathbf{W}_{\text{input}}^{[k]} - \mathbf{W}_{\text{bool}}^{[k]} \odot \left( \mathbf{s}_{\text{out}}^{[k]} \mathbf{s}_{\text{in}}^{[k]\top} \right). \tag{8}$$

Here, $\mathbf{W}_{\text{res}}^{[k]}$ is the residual matrix, and $\mathbf{W}_{\text{bool}}^{[k]}$, $\mathbf{s}_{\text{out}}^{[k]}$ and $\mathbf{s}_{\text{in}}^{[k]}$ are the extracted parameters for the $k$-th kernel, while $\mathbf{W}_{\text{input}}^{[k]}$ represents the input FP matrix for step $k$. For the first step, this is the original weight matrix, and for subsequent steps, it is the residual matrix obtained from the previous step.

Fig. 4 illustrate this process. Although using multiple kernels effectively captures the original weight matrix, a residual error still remains at the end of the process. While this residual error is small, it can accumulate as it propagates through the layers, finally leading to predictions that diverge from those of the original FP model. To address this issue, it is necessary to further finetune the resulting model to compensate for these errors and make it better suited to the target task. We will discuss this in § 6.1.2. In the following section, we will introduce knowledge distillation to achieve this goal.

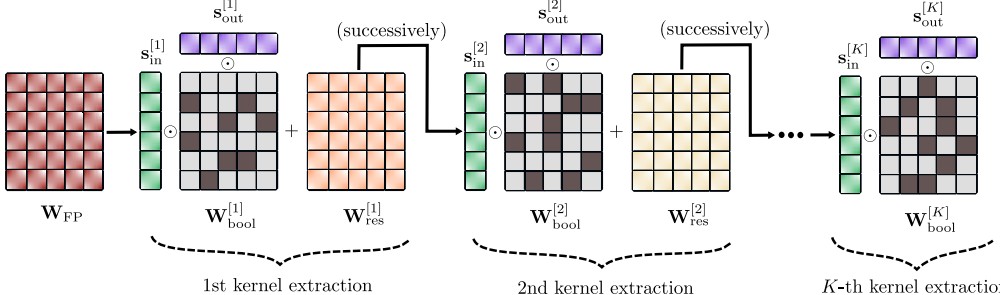

Figure 4: Illustration of successive extractions of Boolean kernels from a given FP weight matrix.

#### 4.3.2 FINETUNING WITH KNOWLEDGE DISTILLATION

Knowledge distillation (KD) (Hinton et al., 2015) trains a student network to mimic a more powerful teacher, usually with greater efficiency. The student learns from the teacher's output distribution and/or intermediate states as "soft targets". Here, the FP model is the teacher and the Boolean model the student. Specifically, the output probability distribution of an LLM for a token $\mathbf{X}_{[i]}$ is:

$$p(\mathbf{X}_{[i]}; \tau) = \frac{\exp(\mathbf{X}_{[i]}/\tau)}{\sum_{j=1}^{N_V} \exp(\mathbf{X}_{[j]}/\tau)}, \tag{9}$$

where $N_V$ is the vocabulary size and $\tau$ is the softmax temperature. The logit-based knowledge distillation (KD) loss across the sequence of all output tokens is defined as follows:

$$\mathcal{L}_{\text{logits}} = \frac{1}{L} \sum_{j=1}^{L} D_{\text{logits}} \left( p_{\text{FP}}(\mathbf{X}_{[j]}; \tau), p_{\text{bool}}(\mathbf{X}_{[j]}; \tau) \right). \tag{10}$$

Here, $p_{\text{FP}}(\mathbf{X}_{[j]}; \tau)$ and $p_{\text{bool}}(\mathbf{X}_{[j]}; \tau)$ denote the distributions over the $j$-th token from the FP and Boolean models, respectively, with $L$ as the sequence length. We find that $\tau = 1$ works best in practice. Among possible measures for $D_{\text{logits}}$ (Ko et al., 2024), the forward Kullback–Leibler (KL) divergence gives the strongest results; further discussion is in Appendix G.2.

To further reduce distributional discrepancies in intermediate layers, we additionally employ an intermediate state–based KD loss over a sequence of hidden states:

$$\mathcal{L}_{\text{is}} = \frac{1}{L} \sum_{h \in H} \sum_{j=1}^{L} \left\| \mathbf{Q}_{\text{FP}}^{j,h} - \mathbf{Q}_{\text{bool}}^{j,h} \right\|_2^2, \tag{11}$$

where $\mathbf{Q}_{\text{FP}}^{j,h}$ and $\mathbf{Q}_{\text{bool}}^{j,h}$ represent the $h$-th hidden states of the FP and Boolean models for the $j$-th token, repsectively; $H$ is the set of chosen intermediate states. Finally, the overall loss is then computed as $\mathcal{L} = \mathcal{L}_{\text{logits}} + \gamma \mathcal{L}_{\text{is}}$, where $\gamma$ is a weighted factor that balances the contribution of the two losses. We empirically found that $\gamma = 10$ works best.

## 5 KERNEL ALLOCATION

Using more kernels enhances the Boolean model's representational capacity but also increases its size. We propose a method to automatically allocate kernels per weight under a fixed budget. Let $N_{\mathbf{W}}$ be the number of weights in the FP teacher model, and $K_l$ for $l \in [1, N_{\mathbf{W}}]$ the number of Boolean kernels for the $l$-th weight. Our goal is to determine $\mathbf{k} \triangleq \{K_l\}_{l \in [1, N_{\mathbf{W}}]}$ subject to design constraints. Key factors include:

*(1) Residual error*: Let $e_l^{[k]} \in \mathbb{R}$ denote the approximation error from applying the successive SVID extraction to the $k$-th kernel of the $l$-th weight, measured by the Frobenius norm of $\mathbf{W}_{\text{res}}^{[k]}$ (Eq. 8).

*(2) Weight importance*: Let $h_l$ denote the importance of the $l$-th weight in the FP teacher model. Higher scores indicate the need for more Boolean kernels. We propose estimating $h_l$ using projection weighted canonical correlation analysis (PWCCA) (Morcos et al., 2018), a reliable method for analyzing deep model representations. Details are provided in Appendix E.1.

*(3) Weight size*: The size of the $l$-th weight is denoted by $s_l$ and $p_l \triangleq s_l / \sum_{k=1}^{N_{\mathbf{W}}} s_k$ represents its relative size in the model.

For a given $\mathbf{k}$, the size of the target Boolean model, in terms of the number of weights, is $\sum_{l=1}^{N_{\mathbf{W}}} K_l s_l$. Relative to the source FP model, this repersents an expansion ratio, defined as:

$$\rho(\mathbf{k}) \triangleq \frac{\sum_{l=1}^{N_{\mathbf{W}}} K_l s_l}{\sum_{l=1}^{N_{\mathbf{W}}} s_l} = \sum_{l=1}^{N_{\mathbf{W}}} K_l p_l. \tag{12}$$

**Optimization objective.** To control model size, we constrain the expansion ratio to a target $T \geq 1$ and limit the kernel size by $K_{\max}$, with $T \leq K_{\max} \leq \infty$. The optimization space is thus $\mathcal{K} \triangleq [1, K_{\max}]^{N_{\mathbf{W}}}$, and the problem is formulated as:

$$\mathbf{k}^* = \arg\min_{\mathbf{k} \in \mathcal{K}} \mathcal{E}(\mathbf{k}), \quad \text{s.t.} \quad \rho(\mathbf{k}) \leq T, \quad \text{where } \mathcal{E}(\mathbf{k}) \triangleq \sum_{l=1}^{N_{\mathbf{W}}} h_l e_l^{[K_l]} f(p_l). \tag{13}$$

Here, $\mathcal{E}(\mathbf{k})$ is the objective (energy) function, and $f(\cdot)$ is a monotonically decreasing function. In practice, we use $f(p_l) = (1/p_l) \log(1/p_l)$. Intuitively, the goal is to minimize residual error while prioritizing weights with higher importance and smaller size, balancing accurate knowledge transfer with model efficiency.

**Optimization algorithm.** The problem has complexity $\mathcal{O}(K_{\max}^{N_{\mathbf{W}}})$, which is prohibitive for LLMs. To tackle this NP-hard problem efficiently, we note that $e_l^{[k]}$ decreases with $k$ for all $l$, and $\mathcal{E}(\mathbf{k})$ is maximized at $\mathbf{k} = \mathbf{1}$, with any increase in $k_l$ reducing $\mathcal{E}(\mathbf{k})$. This motivates a heuristic iterative approach: at each step, increment the $K_l$ that yields the largest reduction in $\mathcal{E}(\mathbf{k})$. The full algorithm is given in Algorithm 9 in the Appendix. We will demonstrate in § 6.5 the practicality of our method.

## 6 EXPERIMENTS

**Setups.** In all experiments, we follow the protocol from Jo et al. (2024), without quantizing activations. The training set combines WikiText2 (Merity et al., 2017) and a selected partition of C4 (Raffel et al., 2020) data, using sequences of length 2048. We apply a cosine decay learning rate with a 3% warm-up over 3 epochs and batch size 8. Boolean parameters use a maximum learning rate of $5 \times 10^{-3}$, while remaining FP parameters are optimized with AdamW (Loshchilov & Hutter, 2019) at a maximum learning rate of $2 \times 10^{-5}$, with $\beta_1 = 0.9$ and $\beta_2 = 0.999$. Following standard practice (Jo et al., 2024), performance is evaluated via perplexity on WikiText2 and C4 (lower is better).

### 6.1 ABLATION STUDIES AND ANALYSIS

#### 6.1.1 EFFECT OF THE NUMBER OF KERNELS

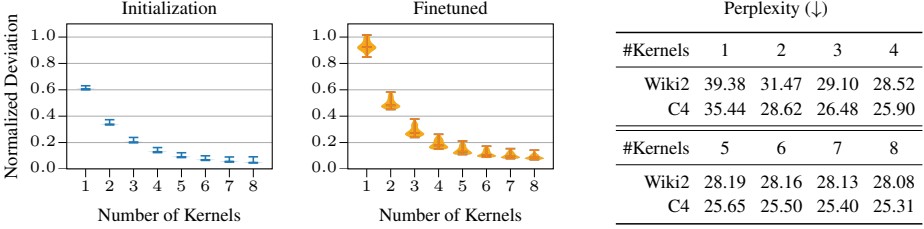

Figure 5: Normalized L1 norm difference between the approximated weights at initialization and after finetuning against the FP weights ($\|\mathbf{W}_{\text{approx}} - \mathbf{W}_{\text{FP}}\|_1 / \|\mathbf{W}_{\text{FP}}\|_1$), and the final results.

We begin by examining the effect of the number of Boolean kernels on OPT-125M model (Zhang et al., 2022). Fig. 5 shows the normalized difference between weights approximated via successive SVID and the original FP weights, both at initialization and after finetuning. Increasing the number of kernels reduces approximation error and improves perplexity, unlike MoS (Jo et al., 2024), where adding more experts does not always help and can even hurt performance. Using 3–4 kernels yields a good approximation, with diminishing improvements beyond that. Interestingly, the normalized difference relative to the full FP weights is larger after KD finetuning. We hypothesize that KD compensates the errors due to the lower expressiveness of a small number of kernels, further emphasizing its role in adapting the model to approximate the FP model rather than exactly replicating each weight.

#### 6.1.2 OPTIMIZATION STRATEGY

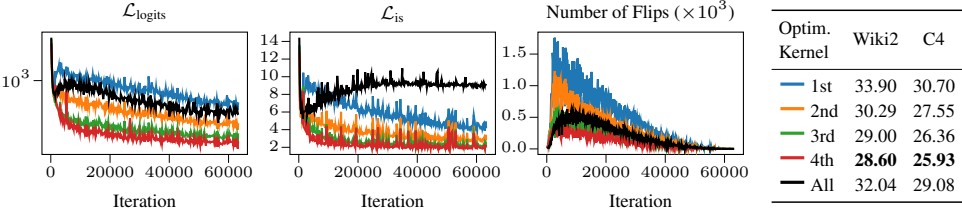

Figure 6: The progression of training losses, number of flips, and perplexity of the resulting models (OPT-125M) is examined with respect to the optimization of different kernel configurations.

Next, we study the effect of optimizing kernels on the OPT-125M model. We consider four Boolean kernels but train only one at a time, keeping the others frozen. Fig. 6 shows the loss convergence. Training the first kernel converges slowest, while higher-order kernels improve progressively. As shown in Proposition 4.1 and Proposition 4.3, the SVID effectively extracts optimal Boolean weights and scaling factors. In our successive SVID framework, the first kernel is well extracted and captures the most important information, and higher-order kernels approximate residuals. Since the kernels are related in a successive manner, modifying lower-order kernels affects higher-order ones. We observe that training only the first kernel results in many weight flips, indicating optimization difficulty, whereas fine-tuning only the last kernel efficiently compensates for residual errors, showing the lowest flip rates and best performance. This is in line with the observation by Liu et al. (2024b), where they compress "delta" induced by the finetuning process by using 1-bit weights. This further

highlights the advantage of our approach, as training complexity is significantly reduced by only optimizing the last kernel. Thus, we apply this strategy in all our experiments.

## 6.2 MAIN BENCHMARK RESULTS

Table 2 compares our method with recent baselines in binarization and 2-bit quantization, evaluating perplexity and accuracy on zero-shot tasks including Winogrande (Sakaguchi et al., 2021), HellaSwag (Zellers et al., 2019), PIQA (Bisk et al., 2020), BoolQ (Clark et al., 2019), and ARC (Clark et al., 2018). For our method, we use 2 Boolean kernels, an ultra low-bit setting. Due to space constraints, the results for LLaMA2-7B and LLaMA2-13B (Touvron et al., 2023b) and different number of Boolean kernels are provided in Appendix G.4 and Appendix G.3. We note that our method is close to scalar quantization while being completely orthogonal to vector quantization (VQ) which adds substantial overhead (Gray, 1984). For completeness, we encourage the reader refer to Appendix G.11 for VQ comparisons, and Appendix G.6 for further baselines.

Our method consistently and significantly outperforms the baselines in both perplexity and zero-shot accuracy, achieving results close to the FP16 baseline despite using only a budget of 2 bits for weight. As expected, QAT methods like OneBit and MoS perform better than PTQ methods, but this comes at the cost of extensive finetuning. In contrast, our approach efficiently address this problem by optimizing parameters directly in Boolean space, avoiding the need for optimizing in FP latent sapce.

Table 2: Perplexity and zero-shot accuracy results of Float16, quantized and binarized LLMs.

| Model | Method | Wbits | Perplexity (↓) | | Zero-shot Accuracy (↑) | | | | | | |
| | | | Wiki2 | C4 | BoolQ | PIQA | Hella. | WinoG. | ARC-e | ARC-c | Average |
|---|---|---|---|---|---|---|---|---|---|---|---|
| OPT-1.3B | FP16 | 16 | 14.62 | 14.72 | 57.82 | 72.42 | 53.70 | 59.51 | 50.97 | 29.52 | 53.99 |
| | PB-LLM | 1.7 | 272.83 | 175.42 | 62.17 | 54.24 | 27.25 | 50.27 | 27.98 | 23.72 | 40.94 |
| | BiLLM | 1.11 | 69.45 | 63.92 | 61.92 | 59.52 | 33.81 | 49.32 | 34.38 | 22.35 | 43.55 |
| | OneBit | 1 | 20.36 | 20.76 | 57.85 | 66.53 | 39.21 | 54.61 | 42.80 | 23.97 | 47.50 |
| | MoS | 1 | 18.45 | 18.83 | 60.34 | 68.66 | 41.99 | 53.99 | 44.87 | 26.19 | 49.34 |
| | OPTQ | 2 | 9.5e3 | 3.8e3 | 39.60 | 52.07 | 25.57 | 49.33 | 26.68 | 23.63 | 35.15 |
| | LLM-QAT | 2 | 4.9e3 | 2.1e3 | 37.83 | 50.05 | 25.72 | 49.72 | 25.76 | 25.09 | 34.07 |
| | OmniQuant | 2 | 42.43 | 55.64 | 56.45 | 60.94 | 33.39 | 51.85 | 38.76 | 23.38 | 44.13 |
| | MBOK [Ours] | 2×1 | **16.13** | **16.61** | 58.53 | 70.67 | 48.11 | 56.75 | 48.19 | 27.90 | **51.69** |
| LLaMA-7B | FP16 | 16 | 5.68 | 7.08 | 73.21 | 77.42 | 72.99 | 66.85 | 52.53 | 41.38 | 64.06 |
| | PB-LLM | 1.7 | 198.37 | 157.35 | 60.51 | 53.53 | 27.23 | 49.17 | 27.48 | 26.02 | 40.66 |
| | BiLLM | 1.11 | 41.66 | 48.15 | 62.23 | 58.65 | 34.64 | 51.14 | 33.08 | 25.68 | 44.24 |
| | OneBit | 1 | 8.48 | 10.49 | 62.50 | 70.40 | 54.03 | 55.32 | 41.07 | 30.88 | 52.36 |
| | MoS | 1 | 7.97 | 9.72 | 64.59 | 71.82 | 58.18 | 58.88 | 42.09 | 31.31 | 54.48 |
| | OPTQ | 2 | 1.9e3 | 7.8e2 | 43.79 | 49.95 | 25.63 | 49.41 | 25.84 | 27.47 | 37.02 |
| | LLM-QAT | 2 | 7.1e2 | 3.0e2 | 37.83 | 50.87 | 24.76 | 51.78 | 26.26 | 25.51 | 36.17 |
| | OmniQuant | 2 | 15.34 | 26.21 | 58.69 | 62.79 | 43.68 | 52.96 | 41.54 | 29.35 | 48.17 |
| | MBOK [Ours] | 2×1 | **6.83** | **8.53** | 69.20 | 74.32 | 64.80 | 60.30 | 49.05 | 34.90 | **58.76** |
| LLaMA-13B | FP16 | 16 | 5.09 | 6.61 | 68.47 | 79.05 | 76.24 | 70.17 | 59.85 | 44.54 | 66.39 |
| | PB-LLM | 1.7 | 35.83 | 39.79 | 62.17 | 58.70 | 33.97 | 52.17 | 31.86 | 23.63 | 43.75 |
| | BiLLM | 1.11 | 14.56 | 16.67 | 62.53 | 68.17 | 52.24 | 59.43 | 41.91 | 29.94 | 52.37 |
| | OneBit | 1 | 7.65 | 9.56 | 63.30 | 71.98 | 60.61 | 59.43 | 42.85 | 32.42 | 55.10 |
| | MoS | 1 | 7.16 | 8.81 | 63.82 | 73.88 | 64.05 | 60.93 | 44.28 | 33.11 | 56.68 |
| | OPTQ | 2 | 3.2e3 | 9.9e2 | 42.39 | 50.00 | 25.27 | 50.67 | 26.14 | 27.39 | 36.98 |
| | LLM-QAT | 2 | 1.8e3 | 1.2e3 | 37.83 | 50.33 | 25.40 | 51.62 | 27.02 | 26.87 | 36.51 |
| | OmniQuant | 2 | 13.43 | 19.33 | 62.20 | 68.99 | 54.16 | 53.83 | 45.50 | 30.38 | 52.51 |
| | MBOK [Ours] | 2×1 | **6.17** | **7.88** | 68.10 | 76.33 | 69.88 | 64.17 | 52.34 | 37.88 | **61.45** |

## 6.3 ACCURACY-COMPRESSION TRADE-OFFS

We further investigate the accuracy-compression trade-offs of our method, quantization methods, and the FP model. Specifically, we compare 3-bit quantization using round-to-nearest (RTN) (Yao et al., 2022; Dettmers et al., 2022) and OPTQ (Frantar et al., 2023) methods against our approach using 3 Boolean kernels. We evaluate these methods on OPT models of varying sizes. The results, presented in Table 3 and Fig. 1, show that with 3 kernels, our method closely approaches the performance of the FP model. Given the same weight budget, our method clearly sits on the Pareto frontier, delivering the best performance for the same model size.

Table 3: OPT perplexity results (*lower is better*) on WikiText2 and C4. The results of FP, round-to-nearest (RTN) and OPTQ are taken from (Frantar et al., 2023).

| OPT Model | WBits | Wiki2 | | | | | C4 | | | | |
|---|---|---|---|---|---|---|---|---|---|---|---|
| | | 125M | 350M | 1.3B | 2.7B | 6.7B | 125M | 350M | 1.3B | 2.7B | 6.7B |
| FULL-PRECISION | 16 | 27.65 | 22.00 | 14.63 | 12.47 | 10.86 | 26.56 | 22.59 | 16.07 | 14.34 | 12.71 |
| RTN (Yao et al., 2022; Dettmers et al., 2022) | 3 | 1.3e3 | 64.57 | 1.3e4 | 1.6e4 | 5.8e3 | 834 | 55.49 | 5.2e3 | 1.1e4 | 5.3e3 |
| OPTQ (Frantar et al., 2023) | 3 | 53.85 | 33.79 | 20.97 | 16.88 | 14.86 | 42.41 | 31.33 | 21.63 | 18.17 | 17.14 |
| MBOK [Ours] | 3×1 | **29.10** | **23.12** | **15.30** | **13.09** | **11.03** | **28.62** | **22.10** | **15.68** | **14.00** | **12.33** |

## 6.4 COMPARISON WITH LATENT-WEIGHT APPROACHES

We compare our method with latent-weight approaches on OPT models, using MoS with 3 experts and our method with 3 Boolean kernels. We also introduce a baseline using our SVID framework to construct 3 binary weights that rely on FP latent weights for training. Results in Fig. 7 show that our method converges much faster, as it directly optimizes Boolean parameters without the need for STE to approximate gradient signals. Both our approach and the latent-weight method outperform MoS, demonstrating the benefit of using additional Boolean kernels and our successive SVID framework. Our method is also more efficient, avoiding the need for FP latent weights and extra momentum.

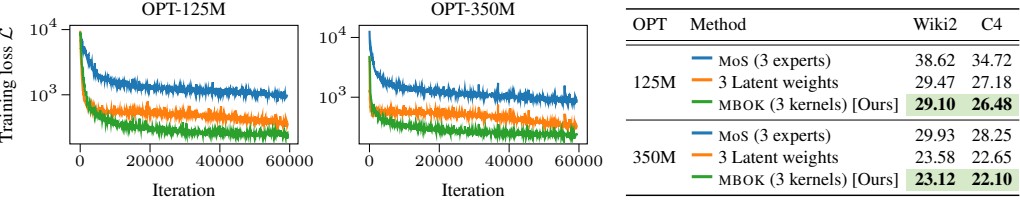

Figure 7: Comparions between our method and latent-weight approaches.

## 6.5 KERNEL ALLOCATION AND COMPARISON TO BITNET B1.58

We next evaluate our kernel allocation method on the OPT-125M model. It supports bit allocation at any granularity, including fractional averages, providing practitioners with a flexible model selection tool under deployment constraints. Fig. 10 reports results for varying average bit budgets, showing consistent improvements as the budget increases. Fig. 9 illustrates kernel allocation with a 3.5-bit average, where more kernels are assigned to FC2 and output projection layers in the final blocks. This aligns with prior observations (Bondarenko et al., 2023; Frantar et al., 2023) that these layers are particularly important and sensitive to compression.

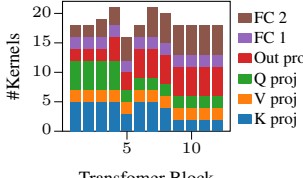

Figure 9: Allocated kernels for OPT-125M.

In addition, our framework's flexibility enables direct comparison with BitNet-b1.58 (Ma et al., 2024), which employs ternary weights. With a 1.58-bit budget, our model achieves reasonable results, whereas BitNet-b1.58 reaches a C4 perplexity of 10199.89 due to finetuning instability, consistent with Xu et al. (2024). We also compare against ShiftAddLLM (You et al., 2024), a PTQ method supporting bit allocation. Our approach performs substantially better (32.23 with a 2-bit budget vs. 435.84 for their mixed 2.2-bit allocation, see Table 17 in ShiftAddLLM).

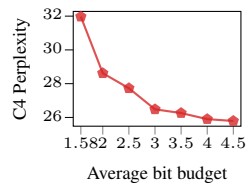

Figure 10: OPT-125M performance w.r.t. bit budget.

## 6.6 DISCUSSION ON COMPLEXITY AND LATENCY

We emphasize the efficiency of our method during finetuning by comparing MoS (Jo et al., 2024) with our approach using 3 Boolean kernels on the OPT-6.7B model. Because we optimize directly in the Boolean domain, each weight requires only 1 bit, whereas MoS relies on 16-bit latent weights. Moreover, we finetune only the last Boolean kernel, with the optimizer storing a single 16-bit momentum per weight. In contrast, Adam (Kingma & Ba, 2015) for latent weights needs two 16-bit momenta per weight. Fig. 11 shows the estimated

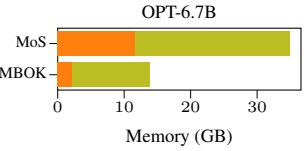

Figure 11: Estimated memory for finetuning for weights ( ■ ) and optimizer states ( ■ ).

memory for a minibatch of one, highlighting the substantial memory savings of our method. These gains could be further amplified by incorporating optimizer state compression techniques such as GaLore (Zhao et al., 2024).

Beyond our theoretical analysis of training complexity (see Appendix F), we provide empirical evidence of practical GPU latency gains (see Appendix G.11 for detailed analyses). Leveraging the BitBLAS library (Wang et al., 2024) for 1-bit matrix multiplications (INT1 weights, FP16 activations) on an A100 GPU, MBOK achieves up to an $8.7\times$ speedup for LLaMA-13B linear layers compared to FP16 baselines at a batch size of 1 (Table 4). Our method significantly outpaces existing binarization and scalar quantization techniques, and runs much faster than SOTA 2-bit vector quantization baselines like QUIP# (Tseng et al., 2024a) and QTIP (Tseng et al., 2024b) while delivering comparable performance. Taken together, our native Boolean approach is a highly efficient alternative to VQ methods, with even greater performance expected on dedicated Boolean hardware.

Table 4: Measured latency (ms) of linear layers in LLaMA-13B, with values in parentheses denoting speed-up relative to the FP16 baseline.

| WEIGHT SIZE | FP16 | QUIP# (Tseng et al., 2024a) | QTIP (Tseng et al., 2024b) | MBOK (Ours) |
|---|---|---|---|---|
| $5120 \times 5120$ | 0.16540 | 0.62260 (0.27×) | 1.96368 (0.08×) | **0.05074 (3.25×)** |
| $5120 \times 13824$ | 0.42830 | 0.62836 (0.68×) | 5.23681 (0.09×) | **0.05098 (8.40×)** |
| $13824 \times 5120$ | 0.43411 | 0.62840 (0.69×) | 5.21193 (0.08×) | **0.04987 (8.70×)** |

## 7 CONCLUSIONS

We introduced Multiple Boolean Kernels (MBOK), a novel framework for low-bit finetuning LLMs. By utilizing Boolean weights and optimizing them directly in the Boolean domain, our framework significantly reduces both memory and computation costs during *both* finetuning and inference. The flexible multi-Boolean structure, along with the proposed successive SVID, effectively transfers knowledge from a source FP model. Through extensive experiments on LLMs of various sizes, we demonstrate that our method approaches FP performance while achieving the best accuracy-compression trade-off compared to existing quantization and binarization methods.

**Limitations.** Our method, like other binarized neural networks, could not be assessed on native Boolean accelerators due to hardware being optimized for real arithmetic. Nevertheless, we demonstrated strong results even on modern hardware, underscoring the promise of our approach and motivating future development of accelerators tailored to Boolean computation.

## ACKNOWLEDGEMENTS

We thank Jean-Claude Belfiore and Li Rong for their feedback during patent review meetings, and Yun Yaw Chu for providing infrastructure support.

## AUTHOR CONTRIBUTIONS STATEMENT

List of Authors: Ba-Hien Tran (B.H.T.), Van Minh Nguyen (V.M.N.).

B.H.T. initiated the low-complexity multi-kernel research direction, defined the narrative of the paper, formulated the core algorithm, proved the mathematical optimality of successive kernel extraction, and proposed both the efficient optimization strategies and the method for estimating weight importance. B.H.T. also designed and conducted all experiments. V.M.N. proposed the heuristic algorithm for kernel allocation. Both authors contributed to the writing and editing of the manuscript, focusing on their respective areas of contribution.

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

# Appendix

## TABLE OF CONTENTS

## A    PRIMER ON BOOLEAN NEURAL NETWORKS

For completeness, this section reviews the concepts and methodology of Boolean neural networks as proposed by Nguyen (2023); Nguyen et al. (2024).

### A.1    NEURON DESIGN

**Boolean Neuron.**    Consider the $l$-th Boolean linear layer; in the forward pass, the output of the next layer is defined as Nguyen et al. (2024):

$$\mathbf{Y}^{(l)}_{[k,j]} = \mathbf{b}^{(l)}_{[j]} + \sum_{i=1}^{n} \mathrm{L}(\mathbf{X}^{(l),\mathbf{W}^{(l)}_{[i,j]}}_{[k,i]}), \qquad 1 \leq j \leq m, \tag{14}$$

where $k$ denotes the sample index in the batch, and $\mathrm{L}$ is a logic gate such as **and**, **or**, **xor**, or **xnor**; The weights $\mathbf{W}^{(l)}_{[i,j]}$ are Boolean values $\{\text{TRUE}, \text{FALSE}\}$ or $\{-1, +1\}$, as typically used in practical implementations. $n$ and $m$ are the number of input and output neurons, respectively. As the most extreme use case, the input data are also Boolean values. The above summation is understood as the counting of TRUE values. We emphasize that the framework is flexible, as it allows Boolean linear layers to be connected through activation layers, layer normalization, arithmetic layers, or other types of layers.

**Mixed Boolean-Real Neuron.**    To enable flexible integration and coexistence of Boolean designs with real-valued components in deep models, we consider two cases of mixed-type data: (i) Boolean weights with real-valued inputs, and (ii) real-valued weights with Boolean inputs. This paper focuses on the first case. These scenarios are addressed through an extension of Boolean logic to accommodate mixed-type data. To proceed, we introduce the essential notations and definitions. Specifically, we define $\mathbb{B} \triangleq \{\text{TRUE}, \text{FALSE}\}$ as the Boolean domain, equipped with standard Boolean logic operations.

> **Definition A.1** (Three-valued logic)**.** *We define the mixed logic domain as $\mathbb{M} \triangleq \mathbb{B} \cup \{0\}$, where $0$ represents an undefined or neutral value. The logic connectives in $\mathbb{M}$ are defined in alignment with standard Boolean logic, as follows. First, the negation operator is extended as: $\neg\text{TRUE} = \text{FALSE}$, $\neg\text{FALSE} = \text{TRUE}$, and $\neg 0 = 0$. Next, let $\mathrm{L}$ denote a generic logic connective (e.g., AND, OR). We distinguish its use in $\mathbb{M}$ and $\mathbb{B}$ by writing $\mathrm{L}_{\mathbb{M}}$ and $\mathrm{L}_{\mathbb{B}}$, respectively. The extended connective $\mathrm{L}_{\mathbb{M}}$ is defined by:*
> $$\mathrm{L}_{\mathbb{M}}(a,b) = \begin{cases} \mathrm{L}_{\mathbb{B}}(a,b) & \text{for } a,b \in \mathbb{B}, \\ 0 & \text{otherwise.} \end{cases}$$

*Notation A.2.*  Denote by $\mathbb{L}$ a logic set (e.g., $\mathbb{B}$ or $\mathbb{M}$), $\mathbb{R}$ the real set, $\mathbb{Z}$ the set of integers, $\mathbb{N}$ a numeric set (e.g., $\mathbb{R}$ or $\mathbb{Z}$), and $\mathbb{D}$ a certain set of $\mathbb{L}$ or $\mathbb{N}$.

> **Definition A.3.** *For $x \in \mathbb{N}$, its logic value denoted by $x_{\text{logic}}$ is given as $x_{\text{logic}} = \text{TRUE} \Leftrightarrow x > 0$, $x_{\text{logic}} = \text{FALSE} \Leftrightarrow x < 0$, and $x_{\text{logic}} = 0 \Leftrightarrow x = 0$.*

> **Definition A.4.** *The magnitude of a variable $x$, denoted by $|x|$, is defined as follows. If $x \in \mathbb{N}$, then $|x|$ is the standard absolute value. For $x \in \mathbb{L}$, the magnitude is given by:*
> $$|x| = \begin{cases} 0 & \text{if } x = 0, \\ 1 & \text{otherwise.} \end{cases}$$

> **Definition A.5** (Mixed-type logic)**.** *For $\mathrm{L}$ a logic connective of $\mathbb{L}$ and variables $a$, $b$, operation $c = \mathrm{L}(a,b)$ is defined such that $|c| = |a||b|$ and $c_{\text{logic}} = \mathrm{L}(a_{\text{logic}}, b_{\text{logic}})$.*

## A.2 MATHEMATICAL FOUNDATION OF BOOLEAN VARIATION

In this section, we present the mathematical foundation of Boolean variation which is the corner stone of the method for training Boolean weights directly within the Boolean domain, without relying on FP latent weights (Nguyen et al., 2024).

### A.2.1 BOOLEAN VARIATION

**Definition A.6.** *Order relations '$<$' and '$>$' in $\mathbb{B}$ are defined as follows:*

$$\text{FALSE} < \text{TRUE}, \quad \text{TRUE} > \text{FALSE}. \tag{15}$$

**Definition A.7.** *For $a, b \in \mathbb{B}$, the variation from $a$ to $b$, denoted $\delta(a \to b)$, is defined as:*

$$\delta(a \to b) \triangleq \begin{cases} \text{TRUE}, & \text{if } b > a, \\ 0, & \text{if } b = a, \\ \text{FALSE}, & \text{if } b < a. \end{cases} \tag{16}$$

**Definition A.8** (Type conversion). *Define:*

$$\mathrm{p} \colon \mathbb{N} \to \mathbb{L}$$

$$x \mapsto \mathrm{p}(x) = \begin{cases} \text{TRUE}, & \text{if } x > 0, \\ 0, & \text{if } x = 0, \\ \text{FALSE}, & \text{if } x < 0. \end{cases} \tag{17}$$

**Proposition A.9.** *(Nguyen, 2023; Nguyen et al., 2024) The following properties hold:*

1. *$\forall x, y \in \mathbb{N}$: $\mathrm{p}(xy) = \mathbf{xnor}(\mathrm{p}(x), \mathrm{p}(y))$.*
2. *$\forall a, b \in \mathbb{L}$: $\mathrm{e}(\mathbf{xnor}(a, b)) = \mathrm{e}(a)\,\mathrm{e}(b)$.*
3. *$\forall x, y \in \mathbb{N}$: $x = y \Leftrightarrow |x| = |y|$ and $\mathrm{p}(x) = \mathrm{p}(y)$.*

In particular, property Proposition A.9(2) implies that by the embedding map $\mathrm{e}(\cdot)$, we have:

$$(\{\text{TRUE}, \text{FALSE}\}, \mathbf{xor}) \cong (\{\pm 1\}, -\times), \tag{18}$$

$$(\{\text{TRUE}, \text{FALSE}\}, \mathbf{xnor}) \cong (\{\pm 1\}, \times), \tag{19}$$

where $\cong$ and $\times$ stand for isomorphic relation, and the real multiplication, resp. A consequence is that by $\mathrm{e}(\cdot)$, a computing sequence of pointwise XOR or XNOR, counting, and majority vote is equivalent to a sequence of pointwise multiplications and accumulation performed on the embedded data.

**Proposition A.10.** *The following properties hold:*

1. *$a \in \mathbb{L}$, $x \in \mathbb{N}$: $\mathbf{xnor}(a, x) = \mathrm{e}(a)x$.*
2. *$x, y \in \mathbb{N}$: $\mathbf{xnor}(x, y) = xy$.*
3. *$x \in \{\mathbb{L}, \mathbb{N}\}$, $y, z \in \mathbb{N}$: $\mathbf{xnor}(x, y + z) = \mathbf{xnor}(x, y) + \mathbf{xnor}(x, z)$.*
4. *$x \in \{\mathbb{L}, \mathbb{N}\}$, $y, \lambda \in \mathbb{N}$: $\mathbf{xnor}(x, \lambda y) = \lambda \mathbf{xnor}(x, y)$.*
5. *$x \in \{\mathbb{L}, \mathbb{N}\}$, $y \in \mathbb{N}$: $\mathbf{xor}(x, y) = -\mathbf{xnor}(x, y)$.*

*Proof.* The proof follows definitions A.5 and A.8.

- Following Definition A.1 we have $\forall t \in \mathbb{M}$, $\mathbf{xnor}(\text{TRUE}, t) = t$, $\mathbf{xnor}(\text{FALSE}, t) = \neg t$, and $\mathbf{xnor}(0, t) = 0$. Put $v = \mathbf{xnor}(a, x)$. We have $|v| = |x|$ and $\mathrm{p}(v) = \mathbf{xnor}(a, \mathrm{p}(x))$. Hence, $a = 0 \Rightarrow \mathrm{p}(v) = 0 \Rightarrow v = 0$; $a = \text{TRUE} \Rightarrow \mathrm{p}(v) = \mathrm{p}(x) \Rightarrow v = x$; $a = \text{FALSE} \Rightarrow \mathrm{p}(v) = \neg \mathrm{p}(x) \Rightarrow v = -x$. Hence (1).

- The result is trivial if $x = 0$ or $y = 0$. For $x, y \neq 0$, put $v = \mathbf{xnor}(x, y)$, we have $|v| = |x||y|$ and $\mathrm{p}(v) = \mathbf{xnor}(\mathrm{p}(x), \mathrm{p}(y))$. According to Definition A.8, if $\mathrm{sign}(x) =$

$\text{sign}(y)$, we have $\text{p}(v) = \text{TRUE} \Rightarrow v = |x||y| = xy$. Otherwise, i.e., $\text{sign}(x) = -\text{sign}(y)$, $\text{p}(v) = \text{FALSE} \Rightarrow v = -|x||y| = xy$. Hence (2).

- (3) and (4) follow (1) for $x \in \mathbb{L}$ and follow (2) for $x \in \mathbb{N}$.

- For (5), write $u = \mathbf{xor}(x, y)$ and $v = \mathbf{xnor}(x, y)$, we have $|u| = |v|$ and $\text{p}(u) = \mathbf{xor}(\text{p}(x), \text{p}(y)) = \neg\mathbf{xnor}(\text{p}(x), \text{p}(y)) = \neg\,\text{p}(v)$. Thus, $\text{sign}(u) = -\text{sign}(v) \Rightarrow u = -v$. $\qquad\square$

*Notation* A.11. We denote $\mathcal{F}(\mathbb{S}, \mathbb{T})$ the set of all functions from source $\mathbb{S}$ to image $\mathbb{T}$.

> ***Definition A.12.*** *For $f \in \mathcal{F}(\mathbb{B}, \mathbb{D})$, $\forall x \in \mathbb{B}$, write $\delta f(x \to \neg x) := \delta(f(x) \to f(\neg x))$. The variation of $f$ w.r.t. $x$, denoted $f'(x)$, is defined as:*
>
> $$f'(x) \triangleq \mathbf{xnor}(\delta(x \to \neg x), \delta f(x \to \neg x)).$$

*Remark* A.13. For convenience and consistency of notation, we intentionally adopt the standard symbol for the continuous derivative, $f'$, to also denote Boolean variation The intended meaning — whether it represents a continuous derivative or a Boolean variation — can be inferred from the context in which the function $f$ is defined. Intuitively, the variation of $f$ w.r.t $x$ is TRUE if $f$ varies in the same direction with $x$.

*Example* A.14. Let $a \in \mathbb{B}$, $f(x) = \mathbf{xor}(x, a)$ for $x \in \mathbb{B}$, the variation of $f$ w.r.t. $x$ can be derived by establishing a truth table (see Table 5) from which we obtain $f'(x) = \neg a$.

Table 5: Variation truth table of $f(x) = \mathbf{xor}(a, x)$, $a, x \in \mathbb{B}$.

| $a$ | $x$ | $\neg x$ | $\delta(x \to \neg x)$ | $f(a, x)$ | $f(a, \neg x)$ | $\delta f(x \to \neg x)$ | $f'(x)$ |
|------|------|------|------|------|------|------|------|
| TRUE | TRUE | FALSE | FALSE | FALSE | TRUE | TRUE | FALSE |
| TRUE | FALSE | TRUE | TRUE | TRUE | FALSE | FALSE | FALSE |
| FALSE | TRUE | FALSE | FALSE | TRUE | FALSE | FALSE | TRUE |
| FALSE | FALSE | TRUE | TRUE | FALSE | TRUE | TRUE | TRUE |

### A.2.2 BOOLEAN VARIATION CALCULUS

Below are some rules of Boolean variation which are necessary for training Boolean neural networks.

> **Proposition A.15.** *(Nguyen, 2023; Nguyen et al., 2024) For $f, g \in \mathcal{F}(\mathbb{B}, \mathbb{B})$, $\forall x, y \in \mathbb{B}$ the following properties hold:*
>
> *1. $\delta f(x \to y) = \mathbf{xnor}(\delta(x \to y), f'(x))$.*
> *2. $(\neg f(x))' = \neg f'(x)$.*
> *3. $(g \circ f)'(x) = \mathbf{xnor}(g'(f(x)), f'(x))$.*

*Proof.* The proof is by definition:

1. $\forall x, y \in \mathbb{B}$, there are two cases. If $y = x$, then the result is trivial. Otherwise, i.e., $y = \neg x$, by definition we have:

$$f'(x) = \mathbf{xnor}(\delta(x \to \neg x), \delta f(x \to \neg x))$$
$$\Leftrightarrow \quad \delta f(x \to \neg x) = \mathbf{xnor}(\delta(x \to \neg x), f'(x)).$$

Hence the result.

2. $\forall x, y \in \mathbb{B}$, it is easy to verify by truth table that $\delta(\neg f(x \to y)) = \neg \delta f(x \to y)$. Hence, by definition,

$$
\begin{aligned}
(\neg f)'(x) &= \mathbf{xnor}(\delta(x \to \neg x), \delta(\neg f(x \to \neg x))) \\
&= \mathbf{xnor}(\delta(x \to \neg x), \neg \delta f(x \to \neg x)) \\
&= \neg \mathbf{xnor}(\delta(x \to \neg x), \delta f(x \to \neg x)) \\
&= \neg f'(x).
\end{aligned}
$$

3. Using definition, property (i), and associativity of $\mathbf{xnor}$, $\forall x \in \mathbb{B}$ we have:

$$
\begin{aligned}
(g \circ f)'(x) &= \mathbf{xnor}(\delta(x \to \neg x), \delta g(f(x) \to f(\neg x))) \\
&= \mathbf{xnor}(\delta(x \to \neg x), \mathbf{xnor}(\delta f(x \to \neg x), g'(f(x)))) \\
&= \mathbf{xnor}(g'(f(x)), \mathbf{xnor}(\delta(x \to \neg x), \delta f(x \to \neg x))) \\
&= \mathbf{xnor}(g'(f(x)), f'(x)).
\end{aligned}
$$

$\square$

> **Proposition A.16.** *(Nguyen, 2023; Nguyen et al., 2024)  For $f \in \mathcal{F}(\mathbb{B}, \mathbb{N})$, the following properties hold:*
> 1. *$x, y \in \mathbb{B}$: $\delta f(x \to y) = \mathbf{xnor}(\delta(x \to y), f'(x))$.*
> 2. *$\alpha \in \mathbb{N}$: $(\alpha f)'(x) = \alpha f'(x)$.*
> 3. *$g \in \mathcal{F}(\mathbb{B}, \mathbb{N})$: $(f + g)'(x) = f'(x) + g'(x)$.*

*Proof.* The proof is as follows:

1. For $x, y \in \mathbb{B}$. Firstly, the result is trivial if $y = x$. For $y \neq x$, i.e., $y = \neg x$, by definition:

$$
f'(x) = \mathbf{xnor}(\delta(x \to \neg x), \delta f(x \to \neg x)).
$$

Hence, $|\delta f(x \to \neg x)| = |f'(x)|$ since $|\delta(x \to \neg x)| = 1$, and

$$
\begin{aligned}
\mathrm{p}(f'(x)) &= \mathbf{xnor}(\delta(x \to \neg x), \mathrm{p}(\delta f(x \to \neg x))) \\
\Leftrightarrow \quad \mathrm{p}(\delta f(x \to \neg x)) &= \mathbf{xnor}(\delta(x \to \neg x), \mathrm{p}(f'(x))),
\end{aligned}
$$

where $\mathrm{p}(\cdot)$ is the logic projector Eq. 17. Thus, $\delta f(x \to \neg x) = \mathbf{xnor}(\delta(x \to \neg x), f'(x))$. Hence the result.

2. Firstly $\forall x, y \in \mathbb{B}$, we have

$$
\delta(\alpha f(x \to y)) = \alpha f(y) - \alpha f(x) = \alpha \delta f(x \to y).
$$

Hence, by definition,

$$
\begin{aligned}
(\alpha f)'(x) &= \mathbf{xnor}(\delta(x \to \neg x), \delta(\alpha f(x \to \neg x))) \\
&= \mathbf{xnor}(\delta(x \to \neg x), \alpha \delta f(x \to \neg x)) \\
&= \alpha \, \mathbf{xnor}(\delta(x \to \neg x), \delta f(x \to \neg x)), \text{ due to Proposition A.10(4)} \\
&= \alpha f'(x).
\end{aligned}
$$

3. For $f, g \in \mathcal{F}(\mathbb{B}, \mathbb{N})$,

$$
\begin{aligned}
(f + g)'(x) &= \mathbf{xnor}(\delta(x \to \neg x), \delta(f + g)(x \to \neg x)) \\
&= \mathbf{xnor}(\delta(x \to \neg x), \delta f(x \to \neg x) + \delta g(x \to \neg x)) \\
&\overset{(*)}{=} \mathbf{xnor}(\delta(x \to \neg x), \delta f(x \to \neg x)) + \mathbf{xnor}(\delta(x \to \neg x), \delta g(x \to \neg x)), \\
&= f'(x) + g'(x),
\end{aligned}
$$

where $(*)$ is due to Proposition A.10(3).

$\square$

For $f \in \mathcal{F}(\mathbb{Z}, \mathbb{N})$, its derivative, also known in terms of *finite differences*, has been defined in the literature as $f'(x) = f(x + 1) - f(x)$, see e.g. Jordan (1950). With the logic variation as introduced above, we can make this definition more generic as follows.

**Definition A.17.** *For $f \in \mathcal{F}(\mathbb{Z}, \mathbb{D})$, the variation of $f$ w.r.t $x \in \mathbb{Z}$ is defined as $f'(x) \triangleq \delta f(x \to x + 1)$, where $\delta f$ is in the sense of the variation defined in $\mathbb{D}$.*

**Proposition A.18.** *(Nguyen, 2023; Nguyen et al., 2024) The following composition rules (chain rules) hold:*

1. *For $\mathbb{B} \xrightarrow{f} \mathbb{B} \xrightarrow{g} \mathbb{D}$: $(g \circ f)'(x) = \mathbf{xnor}(g'(f(x)), f'(x)), \forall x \in \mathbb{B}$.*

2. *For $\mathbb{B} \xrightarrow{f} \mathbb{Z} \xrightarrow{g} \mathbb{D}$, $x \in \mathbb{B}$, if $|f'(x)| \leq 1$ and $g'(f(x)) = g'(f(x) - 1)$, then:*
$$(g \circ f)'(x) = \mathbf{xnor}(g'(f(x)), f'(x)).$$

*Proof.* The proof is as follows.

1. The case of $\mathbb{B} \xrightarrow{f} \mathbb{B} \xrightarrow{g} \mathbb{B}$ is obtained from Proposition A.15(3). For $\mathbb{B} \xrightarrow{f} \mathbb{B} \xrightarrow{g} \mathbb{N}$, by using Proposition A.16(1), the proof is similar to that of Proposition A.15(3).

2. By definition, we have
$$(g \circ f)'(x) = \mathbf{xnor}(\delta(x \to \neg x), \delta g(f(x) \to f(\neg x))). \tag{20}$$
Using property (1) of Proposition A.16, we have:
$$f(\neg x) = f(x) + \delta f(x \to \neg x)$$
$$= f(x) + \mathbf{xnor}(\delta(x \to \neg x), f'(x)). \tag{21}$$
Applying Eq. 21 back to Eq. 20, the result is trivial if $f'(x) = 0$. The remaining case is $|f'(x)| = 1$ for which we have $\mathbf{xnor}(\delta(x \to \neg x), f'(x)) = \pm 1$. First, for $\mathbf{xnor}(\delta(x \to \neg x), f'(x)) = 1$, we have:
$$\delta g(f(x) \to f(\neg x)) = \delta g(f(x) \to f(x) + 1)$$
$$= g'(f(x))$$
$$= \mathbf{xnor}(g'(f(x)), 1)$$
$$= \mathbf{xnor}(g'(f(x)), \mathbf{xnor}(\delta(x \to \neg x), f'(x))). \tag{22}$$
Substitute Eq. 22 back to Eq. 20, we obtain:
$$(g \circ f)'(x) = \mathbf{xnor}(\delta(x \to \neg x), \delta g(f(x) \to f(\neg x)))$$
$$= \mathbf{xnor}(\delta(x \to \neg x), \mathbf{xnor}(g'(f(x)), \mathbf{xnor}(\delta(x \to \neg x), f'(x))))$$
$$= \mathbf{xnor}(g'(f(x)), f'(x)),$$
where that last equality is by the associativity of $\mathbf{xnor}$ and that $\mathbf{xnor}(x, x) = \text{True}$ for $x \in \mathbb{B}$. Similarly, for $\mathbf{xnor}(\delta(x \to \neg x), f'(x)) = -1$, we have:
$$\delta g(f(x) \to f(\neg x)) = \delta g(f(x) \to f(x) - 1)$$
$$= -g'(f(x) - 1)$$
$$= \mathbf{xnor}(g'(f(x) - 1), -1)$$
$$= \mathbf{xnor}(g'(f(x) - 1), \mathbf{xnor}(\delta(x \to \neg x), f'(x))). \tag{23}$$
Substitute Eq. 23 back to Eq. 20 and use the assumption that $g'(f(x)) = g'(f(x) - 1)$, we have:
$$(g \circ f)'(x) = \mathbf{xnor}(\delta(x \to \neg x), \delta g(f(x) \to f(\neg x)))$$
$$= \mathbf{xnor}(\delta(x \to \neg x), \mathbf{xnor}(g'(f(x) - 1), \mathbf{xnor}(\delta(x \to \neg x), f'(x))))$$
$$= \mathbf{xnor}(g'(f(x)), f'(x)).$$
Hence the preposition is proved. □

*Example* A.19. From Example A.14, we have $\delta\mathbf{xor}(x, a)/\delta x = \neg a$ for $a, x \in \mathbb{B}$. Using Proposition A.15-(2) we have: $\delta\mathbf{xnor}(x, a)/\delta x = a$ since $\mathbf{xnor}(x, a) = \neg\mathbf{xor}(x, a)$.

### A.2.3 MULTIVARIATE CASE

The properties of Boolean variation described above can be extended to the multivariate case in a straightforward manner. For example, in the case of multivariate Boolean functions, the extension is as follows.

**Definition A.20.** *For* $\mathbf{x} = (x_1, \ldots, x_n) \in \mathbb{B}^n$, *denote* $\mathbf{x}_{\neg i} \triangleq (x_1, \ldots, x_{i-1}, \neg x_i, x_{i+1}, \ldots, x_n)$ *for* $n \geq 1$ *and* $1 \leq i \leq n$. *For* $f \in \mathcal{F}(\mathbb{B}^n, \mathbb{B})$, *the (partial) variation of* $f$ *w.r.t.* $x_i$, *denoted* $f_i'(\mathbf{x})$ *or* $\delta f(\mathbf{x})/\delta x_i$, *is defined as:* $f_i'(\mathbf{x}) \equiv \delta f(\mathbf{x})/\delta x_i \triangleq \mathbf{xnor}(\delta(x_i \rightarrow \neg x_i), \delta f(\mathbf{x} \rightarrow \mathbf{x}_{\neg i}))$.

The composition rule then becomes:

**Proposition A.21.** *(Nguyen et al., 2024) Let* $f \in \mathcal{F}(\mathbb{B}^n, \mathbb{B})$, $n \geq 1$, *and* $g \in \mathcal{F}(\mathbb{B}, \mathbb{B})$. *For* $1 \leq i \leq n$:
$$(g \circ f)_i'(\mathbf{x}) = \mathbf{xnor}(g'(f(\mathbf{x})), f_i'(\mathbf{x})), \quad \forall \mathbf{x} \in \mathbb{B}^n. \tag{24}$$

*Example* A.22. Apply Proposition A.16-(3) to $\mathbf{Y}_{[k,j]}^{(l)}$ from Eq. 14: $\delta \mathbf{Y}_{[k,j]}^{(l)}/\delta \mathbf{W}_{[i,j]}^{(l)} = \delta \mathrm{L}(\mathbf{X}_{[k,i]}^{(l)}, \mathbf{W}_{[i,j]}^{(l)})/\delta \mathbf{W}_{[i,j]}^{(l)}$ and $\delta \mathbf{Y}_{[k,j]}^{(l)}/\delta \mathbf{X}_{[k,i]}^{(l)} = \delta \mathrm{L}(\mathbf{X}_{[k,i]}^{(l)}, \mathbf{W}_{[i,j]}^{(l)})/\delta \mathbf{X}_{[k,i]}^{(l)}$. Then, for $\mathrm{L} = \mathbf{xnor}$ as an example, we have: $\delta \mathbf{Y}_{[k,j]}^{(l)}/\delta \mathbf{W}_{[i,j]}^{(l)} = \mathbf{X}_{[k,i]}^{(l)}$ and $\delta \mathbf{Y}_{[k,j]}^{(l)}/\delta \mathbf{X}_{[k,i]}^{(l)} = \mathbf{W}_{[i,j]}^{(l)}$.

### A.3 BOOLEAN BACKPROPAGATION

This section presents how to apply the above principles of Boolean variation to define backpropagation for Boolean neural networks. The $l$-th layer (Eq. 14), receives the backpropagation signal from the downstream layer $l + 1$. Specifically, $\mathbf{Z}_{[k,j]}^{(l)} \triangleq \frac{\delta \mathcal{L}}{\delta \mathbf{Y}_{[k,j]}^{(l)}}$ denotes the variation of the loss function $\mathcal{L}$ w.r.t. the output at layer $l$. To optimize the Boolean weights, we need to compute the corresponding loss signal, denoted as $\mathbf{Q}_{[i,j]}^{(l)} \triangleq \frac{\delta \mathcal{L}}{\delta \mathbf{W}_{[i,j]}^{(l)}}$. In addition, we also have to compute the loss signal for the upstream layer, defined as $\mathbf{P}_{[k,i]}^{(l)} \triangleq \frac{\delta \mathcal{L}}{\delta \mathbf{X}_{[k,i]}^{(l)}}$. Hereafter, we consider the logic gate $\mathrm{L} = \mathbf{xnor}$ as a concrete example.

First, using Proposition A.15, Proposition A.16, Proposition A.18 and its extension to the multivariate case by Proposition A.21 in the same manner as shown in Example A.22, we have:

$$\frac{\delta \mathbf{Y}_{[k,j]}^{(l)}}{\delta \mathbf{W}_{[i,j]}^{(l)}} = \frac{\delta \mathbf{xnor}(\mathbf{X}_{[k,i]}^{(l)}, \mathbf{W}_{[i,j]}^{(l)})}{\delta \mathbf{W}_{[i,j]}^{(l)}} = \mathbf{X}_{[k,i]}^{(l)} \tag{25}$$

$$\frac{\delta \mathbf{Y}_{[k,j]}^{(l)}}{\delta \mathbf{X}_{[k,i]}^{(l)}} = \frac{\delta \mathbf{xnor}(\mathbf{X}_{[k,i]}^{(l)}, \mathbf{W}_{[i,j]}^{(l)})}{\delta \mathbf{X}_{[k,i]}^{(l)}} = \mathbf{W}_{[i,j]}^{(l)} \tag{26}$$

Using the chain rules given by Proposition A.18, we have the following atomic variations:

$$\mathbf{Q}_{[k,i,j]}^{(l)} \triangleq \frac{\delta \mathcal{L}}{\delta \mathbf{W}_{[i,j]}^{(l)}}\Big|_k = \mathbf{xnor}\left(\frac{\delta \mathcal{L}}{\delta \mathbf{Y}_{[k,j]}^{(l)}}, \frac{\delta \mathbf{Y}_{[k,j]}^{(l)}}{\delta \mathbf{W}_{[i,j]}^{(l)}}\right) = \mathbf{xnor}\left(\mathbf{Z}_{[k,j]}^{(l)}, \mathbf{X}_{[k,i]}^{(l)}\right), \tag{27}$$

$$\mathbf{P}_{[k,i,j]}^{(l)} \triangleq \frac{\delta \mathcal{L}}{\delta \mathbf{X}_{[k,i]}^{(l)}}\Big|_j = \mathbf{xnor}\left(\frac{\delta \mathcal{L}}{\delta \mathbf{Y}_{[k,j]}^{(l)}}, \frac{\delta \mathbf{Y}_{[k,j]}^{(l)}}{\delta \mathbf{X}_{[k,i]}^{(l)}}\right) = \mathbf{xnor}\left(\mathbf{Z}_{[k,j]}^{(l)}, \mathbf{W}_{[i,j]}^{(l)}\right). \tag{28}$$

The variations $\mathbf{Q}_{[i,j]}^{(l)}$ and $\mathbf{G}_{[k,i]}^{(l)}$ can be then obtained by aggregating the above atomic variations over the batch dimension $k$ and output dimension $j$, respectively. More specifically, denote $\mathbf{1}(\cdot)$ the indicator function. Additionally, for $b \in \mathbb{B}$ and a variable $x$, we define $\mathbf{1}(x = b) = 1$ if $x_{\mathrm{logic}} = b$

and $\mathbf{1}(x = b) = 0$ otherwise. Then, we have:

$$\mathbf{Q}_{[i,j]}^{(l)} \triangleq \frac{\delta\mathcal{L}}{\delta\mathbf{W}_{[i,j]}^{(l)}} = \sum_k \mathbf{1}(\mathbf{Q}_{[k,i,j]}^{(l)} = \text{TRUE})|\mathbf{Q}_{[k,i,j]}^{(l)}| - \sum_k \mathbf{1}(\mathbf{Q}_{[k,i,j]}^{(l)} = \text{FALSE})|\mathbf{Q}_{[k,i,j]}^{(l)}|, \quad (29)$$

$$\mathbf{P}_{[i,j]}^{(l)} \triangleq \frac{\delta\mathcal{L}}{\delta\mathbf{X}_{[k,i]}^{(l)}} = \sum_j \mathbf{1}(\mathbf{P}_{[k,i,j]}^{(l)} = \text{TRUE})|\mathbf{P}_{[k,i,j]}^{(l)}| - \sum_j \mathbf{1}(\mathbf{P}_{[k,i,j]}^{(l)} = \text{FALSE})|\mathbf{P}_{[k,i,j]}^{(l)}|. \quad (30)$$

### A.4 BOOLEAN OPTIMIZER

---

**Algorithm 1:** Boolean learning process for a linear layer.

---
**Input**      : Learning rate $\eta$, number of iterations $T$;

**Initialize :** $\mathbf{M}_{[i,j]}^{(l),0} = 0; \beta^0 = 1$;

1 **for** $t = 0, \ldots, T-1$ **do**

    /* 1.  Forward */

2    Compute $\mathbf{Y}^{(l),t}$ following Eq. 14;

    /* 2.  Backward */

3    Receive $\frac{\delta\mathcal{L}^{(l),t}}{\delta\mathbf{Y}_{[k,j]}^{(l),t}}$ from downstream layer;

    /* 2.1 Backpropagation */

4    Compute and backpropagate $\mathbf{P}^{(l),t}$ to the upstream following Eq. 30;

    /* 2.2 Weight update process */

5    $N_{\text{total}} := 0, N_{\text{unchanged}} := 0$;

6    **foreach** $\mathbf{W}_{i,j}^l$ **do**

7       Compute $\mathbf{Q}_{[i,j]}^{(l),t+1}$ following Eq. 29;

8       Update $\mathbf{M}_{[i,j]}^{(l),t+1} = \beta^t \mathbf{M}_{[i,j]}^{(l),t} + \eta^t \mathbf{Q}_{[i,j]}^{(l),t+1}$;

9       $N_{\text{total}} \leftarrow N_{\text{total}} + 1$;

10      **if** $\text{xnor}(\mathbf{M}_{[i,j]}^{(l),t+1}, \mathbf{W}_{[i,j]}^{(l),t}) = \text{TRUE}$ **then**

        /* Flip weight */

11        $\mathbf{W}_{[i,j]}^{(l),t+1} = \neg\mathbf{W}_{[i,j]}^{(l),t}$;

        /* Reset corresponding accumulator */

12        $\mathbf{M}_{[i,j]}^{(l),t+1} = 0$;

13      **else**

        /* Weight is unchanged */

14        $\mathbf{W}_{[i,j]}^{(l),t+1} = \mathbf{W}_{[i,j]}^{(l),t}$;

        /* Update statistics to update $\beta$ */

15        $N_{\text{unchanged}} \leftarrow N_{\text{unchanged}} + 1$;

16    Update $\eta^{t+1}, \beta^{t+1} = N_{\text{unchanged}}/N_{\text{total}}$ ;

---

Given the above variations, the rule for updating the Boolean weight $\mathbf{W}_{[i,j]}^{(l)}$ to minimize the loss function $\mathcal{L}$ is as follows:

$$\mathbf{W}_{[i,j]}^{(l)} = \neg\mathbf{W}_{[i,j]}^{(l)} \quad \text{if } \text{xnor}\left(\mathbf{Q}_{[i,j]}^{(l)}, \mathbf{W}_{[i,j]}^{(l)}\right) = \text{TRUE}. \quad (31)$$

Based on this update rule, we can develop an optimizer that accumulates the signal $\mathbf{Q}_{[i,j]}^{(l)}$ over training iterations. Specifically, let $\mathbf{W}_{[i,j]}^{(l),t}$ denotes the weight at iteration $t$, and $\mathbf{M}_{[i,j]}^{(l),t}$ represents its accumulator, initialized as $\mathbf{M}_{[i,j]}^{(l),0} = 0$. The update rule for the accumulator is then defined as: The update rule for the accumulator is then defined as:

$$\mathbf{M}_{[i,j]}^{(l),t+1} \leftarrow \beta^t \mathbf{M}_{[i,j]}^{(l),t} + \eta \mathbf{Q}_{[i,j]}^{(l),t}, \quad (32)$$

where $\eta$ is the accumulation factor acting as a learning rate, and $\beta^t$ is an auto-regularizing factor that reflects the system's state at time $t$. In our work, we use brain plasticity (Fuchs et al., 2014)

and Hebbian theory (Hebb, 2005) to adaptively set $\beta^t$, that force the weights to adapt to their neighborhood during. For the chose weight's neighborhood, for instance, neuron, layer, or network level, $\beta^t$ is set as:

$$\beta^t = \frac{\text{Number of unchanged weights at } t}{\text{Total number of weights}}. \tag{33}$$

It to temper the importance of weight variational according to how much neurons have changed. In our experiments, $\beta^t$ is set to per-layer basis and initialized as $\beta^0 = 1$ The learning process for a linear layer is described in Algorithm 1.

## B    DISCUSSION ON HARDWARE CONSIDERATIONS

### B.1    COMPUTATION PROPOSED IN § 4.1

The Boolean framework supports both full and partial binary settings. The afforementioned Boolean variation calculus shows that:

$$\mathbf{xnor}(x_{\text{real}}, w_{\text{logic}}) = x_{\text{real}} \times w_{\text{binary}}, \tag{34}$$

under the mapping TRUE $\to +1$ and FALSE $\to -1$. Consequently, matrix multiplication ($\mathbf{matmul}$) between a real tensor $\mathbf{X}$ and a logic tensor $\mathbf{W}$ can be implemented as follows:

- **Using binary weights** $\{-1, +1\}$: Simply represent the logic weights in binary format. Then, $\mathbf{matmul}(x_{\text{real}}, w_{\text{logic}})$ is directly computed as $\mathbf{matmul}(x_{\text{real}}, w_{\text{binary}})$.

- **Using native logic** $\{\text{TRUE}, \text{FALSE}\}$: The multiplication reduces to:

$$\mathbf{matmul}(x_{\text{real}}, w_{\text{logic}}) = \begin{cases} x_{\text{real}}, & \text{if } w_{\text{logic}} = \text{TRUE} \\ -x_{\text{real}}, & \text{if } w_{\text{logic}} = \text{FALSE} \end{cases} \tag{35}$$

Thus, a sign flip of $x_{\text{real}}$ conditioned on $w_{\text{logic}}$, followed by accumulation, suffices to perform $\mathbf{matmul}(\mathbf{X}_{\text{real}}, \mathbf{W}_{\text{logic}})$.

The first approach is well-supported by modern hardware such as CPUs, GPUs, etc, where different bit-widths can be used to represent and simulate weight values in $\{-1, +1\}$. Additionally, this approach can be implemented directly in PyTorch (Paszke et al., 2019). The second approach, in contrast, requires a specialized Boolean accelerator. Such hardware can massively accelerate the computation by directly leveraging logic operations instead of real-arithmetic.

### B.2    MULTI-CORE COMPUTATION STRATEGY IN § 4.2

Boolean design, as used in the paper, employs Boolean weights and operates using logic operations. It is distinct from bit-level operations.

**Boolean design:**    Weights are Boolean logic variables, taking values TRUE/FALSE or $-1/+1$. Operations are logic-based, such as $\mathbf{xnor}$, and $\mathbf{or}$, etc. See Eq. 35 for an example.

**Bit-level operations:**    These, such as bit-serial implementations in C/C++, operate bit-by-bit on multi-bit variables. For instance, a bit-level AND between two $n$-bit variables produces an $n$-bit result, where each bit is the ADN of corresponding pair of bits from the inputs. Bit-level operations like bit-serial are inefficient in terms of latency, whereas Boolean logic operations are significantly faster compared to real-arithmetic operations such as multiplication.

## C    CODE SAMPLES OF CORE IMPLEMENTATION

### C.1    BOOLEAN LINEAR LAYER AND OPTIMIZER

In this section, we provide example Python code for implementing a Boolean linear layer based on the **xor** logic gate. This implementation is based on the PyTorch framework (Paszke et al., 2019). As done in Nguyen et al. (2024), the class definition for the Boolean linear layer is presented in Algorithm 2, and its backpropagation mechanism—customized via PyTorch's `autograd` system—is detailed in Algorithm 3. Each Boolean kernel is primarily implemented using this Boolean linear layer.

We consider both cases of the incoming backpropagation signal: Boolean-valued (see Algorithm 4), and real-valued (see Algorithm 5). The latter is the main use case in this paper. An example implementation of the Boolean optimizer used to update the layer's parameters is provided in Algorithm 6.

---

**Algorithm 2:** Python code of XOR linear layer

```python
import torch

from torch import Tensor, nn, autograd
from typing import Any, List, Optional, Callable

class XORLinear(nn.Linear):

    def __init__(self, in_features: int, out_features: int, bool_bprop: bool, **kwargs):
        super(XORLinear, self).__init__(in_features, out_features, **kwargs)
        self.bool_bprop = bool_bprop

    def reset_parameters(self):
        self.weight = nn.Parameter(torch.randint(0, 2, self.weight.shape))

        if self.bias is not None:
            self.bias = nn.Parameter(torch.randint(0, 2, (self.out_features,)))

    def forward(self, X):
        return XORFunction.apply(X, self.weight, self.bias, self.bool_bprop)
```

---

**Algorithm 3:** Python code of the backpropagation logic of XOR linear layer

```python
class XORFunction(autograd.Function):

    @staticmethod
    def forward(ctx, X, W, B, bool_bprop: bool):
        ctx.save_for_backward(X,W,B)
        ctx.bool_bprop = bool_bprop

        # Elementwise XOR logic
        S = torch.logical_xor(X[:,None,:], W[None,:,:])

        # Sum over the input dimension
        S = S.sum(dim=2) + B

        # 0-centered for use with BatchNorm when preferred
        S = S - W.shape[1]/2

        return S

    @staticmethod
    def backward(ctx, Z):
        if ctx.bool_bprop:
            G_X, G_W, G_B = backward_bool(ctx, Z)
        else:
            G_X, G_W, G_B = backward_real(ctx, Z)

        return G_X, G_W, G_B, None
```

**Algorithm 4:** Backpropagation logic with Boolean received backpropagation

```
1  def backward_bool(ctx, Z):
2      """
3      Variation of input:
4          - delta(xor(x,w))/delta(x) = neg w
5          - delta(Loss)/delta(x) = xnor(z,neg w) = xor(z,w)
6      Variation of weights:
7          - delta(xor(x,w))/delta(w) = neg x
8          - delta(Loss)/delta(x) = xnor(z,neg x) = xor(z,x)
9      Variation of bias:
10         - bias = xnor(bias,True) ==> Variation of bias is driven in
11           the same basis as that of weight with xnor logic and input True.
12     Aggregation:
13         - Count the number of TRUEs = sum over the Boolean data
14         - Aggr = TRUEs - FALSEs = TRUEs - (TOT - TRUEs) = 2TRUES - TOT
15           where TOT is the size of the aggregated dimension
16     """
17     X, W, B = ctx.saved_tensors
18
19     # Boolean variation of input
20     G_X = torch.logical_xor(Z[:,:,None], W[None,:,:])
21
22     # Aggregate over the out_features dimension
23     G_X = 2 * G_X.sum(dim=1) - W.shape[0]
24
25     # Boolean variation of weights
26     G_W = torch.logical_xor(Z[:,:,None], X[:,None,:])
27
28     # Aggregate over the batch dimension
29     G_W = 2 * G_W.sum(dim=0) - X.shape[0]
30
31     # Boolean variation of bias
32     if B is not None:
33         # Aggregate over the batch dimension
34         G_B = 2 * Z.sum(dim=0) - Z.shape[0]
35
36     # Return
37     return G_X, G_W, G_B
```

**Algorithm 5:** Backpropagation logic with real received backpropagation

```
1  def backward_real(ctx, Z):
2      X, W, B = ctx.saved_tensors
3
4      """
5      Boolean variation of input processed using torch avoiding loop:
6          -> xor(Z: Real, W: Boolean) = -Z * emb(W)
7          -> emb(W): T->1, F->-1 => emb(W) = 2W-1
8          => delta(Loss)/delta(X) = Z*(1-2W) """
9      G_X = Z.mm(1-2*W)
10
11     """
12     Boolean variation of weights processed using torch avoiding loop:
13         -> xor(Z: Real, X: Boolean) = -Z * emb(X)
14         -> emb(X): T->1, F->-1 => emb(X) = 2X-1
15         => delta(Loss)/delta(W) = Z^T * (1-2X) """
16     G_W = Z.t().mm(1-2*X)
17
18     """ Boolean variation of bias """
19     if B is not None:
20         G_B = Z.sum(dim=0)
21
22     # Return
23     return G_X, G_W, G_B
```

---

**Algorithm 6:** Python code of Boolean optimizer

---

```python
class BooleanOptimizer(torch.optim.Optimizer):

    def __init__(self, params, lr: float):
        super(BooleanOptimizer, self).__init__(params, dict(lr=lr))
        for param_group in self.param_groups:
            param_group['accums'] = [torch.zeros_like(p.data) for p in param_group['params']]
            param_group['ratios'] = [0 for p in param_group['params']]
        self._nb_flips = 0

    @property
    def nb_flips(self):
        n = self._nb_flips
        self._nb_flips = 0
        return n

    def step(self):
        for param_group in self.param_groups:
            for idx, p in enumerate(param_group['params']):
                self.update(p, param_group, idx)

    def update(self, param: Tensor, param_group: dict, idx: int):
        accum = param_group['ratios'][idx] * param_group['accums'][idx] + param_group['lr'] * param.grad.data
        param_group['accums'][idx] = accum
        param_to_flip = accum * (2*param.data-1) >= 1
        param.data[param_to_flip] = torch.logical_not(param.data[param_to_flip])
        param_group['accums'][idx][param_to_flip] = 0.
        param_group['ratios'][idx] = 1 - param_to_flip.float().mean()
        self._nb_flips += float(param_to_flip.float().sum())
```

---

## C.2  SUCCESSIVE SVID FOR KERNEL EXTRACTION

Algorithm 7 illustrate the Python code of the SVID algortithm to extract the optimal Boolean weights and scaling factors for one kernel. Based on this, Algorithm 8 illustrates the succesive SVID algorithm to extract all kernels.

---

**Algorithm 7:** Python code of SVID approximation of a FP matrix.

---

```python
def svid_approximation(w):
    """
    Approximate the input matrix 'w' by a boolean matrix and a rank-1 matrix:
        w = w_bool * (s_out * s_in.T)

    Args:
        w (torch.Tensor): Input tensor of shape (*, m, n).

    Returns:
        tuple:
            - w_bool (torch.Tensor): Boolean matrix of the same shape as 'w'.
            - w_res (torch.Tensor): Residual matrix, w - w_bool * (s_out * s_in.T).
            - s_in (torch.Tensor): Scaled first left singular vector of 'w'.
            - s_out (torch.Tensor): Scaled first right singular vector of 'w'.
    """
    U, S, Vh = torch.linalg.svd(abs(w.data.clone().float()), full_matrices=False)

    w_bool = torch.sign(w)
    s_in = torch.sqrt(S[0]) * Vh[0,:].reshape(1,-1)
    s_out = torch.sqrt(S[0]) * U[:,0].reshape(-1,1)

    w_res = w - w_bool * torch.matmul(s_out, s_in)

    return w_bool, w_res, s_in, s_out
```

---

---

**Algorithm 8:** Python code of successively extracts kernels from FP matrix using SVID.

---

```python
def successive_svid(w_fp, n_kernels):
    """
    Perform successive SVID on the input matrix to extract Boolean kernels.

    Args:
        w_fp (torch.Tensor): Input weight matrix.
        n_kernels (int): Number of iterations to extract kernels.

    Returns:
        list: List of dictionaries containing `n_kernels` kernels, each has:
            - w_bool (torch.Tensor): Boolean matrix.
            - s_in (torch.Tensor): Input scaling vector.
            - s_out (torch.Tensor): Output scaling vector.
    """
    boolean_kernels = []

    w = w_fp # The input to SVID at first iteration is the original weight

    for k in range(n_kernels):
        # Extract the Boolean weights, residual, and scaling vectors
        w_bool, w_res, s_in, s_out = svid_approximation(w)

        # Save the extracted kernel
        boolean_kernels.append({'w_bool': w_bool, 's_in': s_in, 's_out': s_out})

        # The input to SVID for the next iteration is the current residual matrix
        w = w_res

    return boolean_kernels
```

---

# D  PROOF OF PROPOSITIONS

For completeness, we include the proofs of Propositions related to SVID approximation used in the main paper.

## D.1  PROOF OF BOOLEAN LINEAR REFORMULATION USING SVID

**Proposition D.1.** *(Xu et al., 2024) Given the weight matrix $\mathbf{W}_{\mathrm{FP}}$ and input $\mathbf{X}$, the linear layer can be reformulated as the following using SVID approximation, $\mathbf{W}_{\mathrm{FP}} \approx \mathbf{W}_{\mathrm{bool}} \odot \left(\mathbf{s}_{\mathrm{out}}\mathbf{s}_{\mathrm{in}}^{\top}\right)$, as follows:*

$$\mathbf{X}\mathbf{W}_{\mathrm{FP}}^{\top} \approx \left[\left(\mathbf{X} \odot \mathbf{s}_{\mathrm{in}}^{\top}\right)\mathbf{W}_{\mathrm{bool}}^{\top}\right] \odot \mathbf{s}_{\mathrm{out}}^{\top}. \tag{36}$$

*Proof.* Due to the SVID approximation, we have $\mathbf{W}_{\mathrm{FP}[i,j]} \approx \mathbf{W}_{\mathrm{bool}[i,j]}\mathbf{s}_{\mathrm{out}[i]}\mathbf{s}_{\mathrm{in}[j]}$. Then, we have:

$$\left(\mathbf{X}\mathbf{W}_{\mathrm{FP}}^{\top}\right)_{[i,j]} \approx \sum_k \mathbf{X}_{[i,k]}\mathbf{W}_{\mathrm{FP}[k,j]}^{\top} \tag{37}$$

$$= \sum_k \mathbf{X}_{[i,k]}\mathbf{W}_{\mathrm{FP}[j,k]} \tag{38}$$

$$= \sum_k \mathbf{X}_{[i,k]}\mathbf{W}_{\mathrm{bool}[j,k]}\mathbf{s}_{\mathrm{out}[j]}\mathbf{s}_{\mathrm{in}[k]} \tag{39}$$

$$= \sum_k \mathbf{X}_{[i,k]}\mathbf{s}_{\mathrm{in}[k]}\mathbf{W}_{\mathrm{bool}[j,k]}\mathbf{s}_{\mathrm{out}[j]} \tag{40}$$

$$= \sum_k \left(\mathbf{X} \odot \mathbf{s}_{\mathrm{in}}^{\top}\right)_{[i,k]}\mathbf{W}_{\mathrm{bool}[k,j]}^{\top}\mathbf{s}_{\mathrm{out}[j]} \tag{41}$$

$$= \left[\left(\mathbf{X} \odot \mathbf{s}_{\mathrm{in}}^{\top}\right)\mathbf{W}_{\mathrm{bool}}^{\top}\right]_{[i,j]}\mathbf{s}_{\mathrm{out}[j]} \tag{42}$$

$$= \left\{\left[\left(\mathbf{X} \odot \mathbf{s}_{\mathrm{in}}^{\top}\right)\mathbf{W}_{\mathrm{bool}}^{\top}\right] \odot \mathbf{s}_{\mathrm{out}}^{\top}\right\}_{[i,j]}. \tag{43}$$

Thus, the proposition is proved. □

## D.2 PROOF OF PROPOSITION 4.1

**Lemma D.2.** *(Xu et al., 2024) Denote $\sigma_i(\mathbf{W})$ the $i$-th biggest singular value of matrix $\mathbf{W}$. The following inequality holds:*

$$\sigma_1(|\mathbf{W}|) \geq \sigma_1(\mathbf{W}). \tag{44}$$

*Proof.* By the definition of induced norm, we have:

$$\sigma_1(\mathbf{W}) = \|\mathbf{W}\|_2 = \max_{\mathbf{x}, \|\mathbf{x}\|_2=1} \|\mathbf{W}\mathbf{x}\|_2, \tag{45}$$

$$\sigma_1(|\mathbf{W}|) = \||\mathbf{W}|\|_2 = \max_{\mathbf{y}, \|\mathbf{y}\|_2=1} \||\mathbf{W}|\mathbf{y}\|_2. \tag{46}$$

In addition, because $\forall \mathbf{x}, \|\mathbf{x}\|_2 = 1$, we have:

$$\||\mathbf{W}||\mathbf{x}|\|_2^2 = \sum_i \left( \sum_j |\mathbf{W}_{[i,j]}||\mathbf{x}_{[j]}| \right)^2 \tag{47}$$

$$\geq \sum_i \left( |\sum_j \mathbf{W}_{[i,j]}\mathbf{x}_{[j]}| \right)^2 \tag{48}$$

$$= \sum_i \left( \sum_j \mathbf{W}_{[i,j]}\mathbf{x}_{[j]} \right)^2 \tag{49}$$

$$= \|\mathbf{W}\mathbf{x}\|_2^2. \tag{50}$$

Therefore

$$\max_{\mathbf{y}, \|\mathbf{y}\|_2=1} \||\mathbf{W}|\mathbf{y}\|_2 \geq \max_{\mathbf{x}, \|\mathbf{x}\|_2=1} \|\mathbf{W}\mathbf{x}\|_2 \tag{51}$$

$$\Leftrightarrow \sigma_1(|\mathbf{W}|) \geq \sigma_1(\mathbf{W}). \tag{52}$$

Thus, the lemma is proved. □

**Proposition D.3** (Restated from Xu et al. (2024)). *For $\mathbf{W} \in \mathbb{R}^{m \times n}$, write $\mathbf{W} = \widetilde{\mathbf{U}}\widetilde{\boldsymbol{\Sigma}}\widetilde{\mathbf{V}}^\top$ its* SVD. *Let $\mathbf{a} = \sqrt{\widetilde{\sigma}_1}\widetilde{\mathbf{U}}_{[:,1]}$, and $\mathbf{b} = \sqrt{\widetilde{\sigma}_1}\widetilde{\mathbf{V}}_{[:,1]}$. Similarly, denote $|\mathbf{W}| = \mathbf{U}\boldsymbol{\Sigma}\mathbf{V}^\top$ its* SVD; $\mathbf{s}_{\mathrm{in}}$ *and $\mathbf{s}_{\mathrm{out}}$ are given as: $\mathbf{s}_{\mathrm{in}} = \sqrt{\sigma_1}\mathbf{V}_{[:,1]}$, and $\mathbf{s}_{\mathrm{out}} = \sqrt{\sigma_1}\mathbf{U}_{[:,1]}$. We decompose the matrix as $\mathbf{W} = \mathbf{W}_{\mathrm{bool}} \odot |\mathbf{W}| \approx \mathbf{W}_{\mathrm{bool}} \odot (\mathbf{s}_{\mathrm{out}}\mathbf{s}_{\mathrm{in}}^\top)$. We then have:*

$$\left\| \mathbf{W} - \mathbf{W}_{\mathrm{bool}} \odot \mathbf{s}_{\mathrm{out}}\mathbf{s}_{\mathrm{in}}^\top \right\|_F^2 \leq \left\| \mathbf{W} - \mathbf{a}\mathbf{b}^\top \right\|_F^2. \tag{53}$$

*Proof.* We denote the following error matrices:

$$\mathbf{E}_1 = \mathbf{W} - \mathbf{a}\mathbf{b}^\top, \tag{54}$$

$$\mathbf{E}_2 = |\mathbf{W}| - \mathbf{s}_{\mathrm{out}}\mathbf{s}_{\mathrm{in}}^\top. \tag{55}$$

Multiplying $\mathbf{W}_{\mathrm{bool}}$ with both sides of Eq. 55, we have:

$$\mathbf{W}_{\mathrm{bool}} \odot |\mathbf{W}| - \mathbf{W}_{\mathrm{bool}} \odot \mathbf{s}_{\mathrm{out}}\mathbf{s}_{\mathrm{in}}^\top = \mathbf{W}_{\mathrm{bool}} \odot \mathbf{E}_2 \tag{56}$$

$$\Leftrightarrow \mathbf{W} - \mathbf{W}_{\mathrm{bool}} \odot \mathbf{s}_{\mathrm{out}}\mathbf{s}_{\mathrm{in}}^\top = \mathbf{W}_{\mathrm{bool}} \odot \mathbf{E}_2. \tag{57}$$

Thus, we have:

$$\|\mathbf{W} - \mathbf{W}_{\text{bool}} \odot \mathbf{s}_{\text{out}}\mathbf{s}_{\text{in}}^{\top}\|_F^2 = \|\mathbf{W}_{\text{bool}} \odot \mathbf{E}_2\|_F^2 \tag{58}$$

$$= \sum_{i,j} \mathbf{W}_{\text{bool}[i,j]}^2 + \mathbf{E}_{2[i,j]}^2 \tag{59}$$

$$= \sum_{i,j} \mathbf{E}_{2[i,j]}^2 \tag{60}$$

$$= \|\mathbf{E}_2\|_F^2 \tag{61}$$

For SVD decomposition, the norm of the above error matrices in the rank-1 approximation is the um of squares of all singular values except the largest one. In particular, we have:

$$\|\mathbf{E}_1\|_F^2 = \sum_{i=2}^{n} \sigma_i^2(\mathbf{W}), \tag{62}$$

$$\|\mathbf{E}_2\|_F^2 = \sum_{i=2}^{n} \sigma_i^2(|\mathbf{W}|). \tag{63}$$

Since $\|\mathbf{W}\|_F^2 = \||\mathbf{W}|\|_F^2$, we have:

$$\sum_{i=1}^{n} \sigma_i^2(\mathbf{W}) = \sum_{i=1}^{n} \sigma_i^2(|\mathbf{W}|) \tag{64}$$

$$\Leftrightarrow \|\mathbf{E}_1\|_F^2 + \sigma_1^2(\mathbf{W}) = \|\mathbf{E}_2\|_F^2 \sigma_1^2(|\mathbf{W}|). \tag{65}$$

Thus, according to Lemma D.2 and Eq. 61, we have:

$$\|\mathbf{E}_2\|_F^2 \leq \|\mathbf{E}_1\|_F^2 \tag{66}$$

$$\left\|\mathbf{W} - \mathbf{W}_{\text{bool}} \odot \mathbf{s}_{\text{out}}\mathbf{s}_{\text{in}}^{\top}\right\|_F^2 \leq \left\|\mathbf{W} - \mathbf{a}\mathbf{b}^{\top}\right\|_F^2. \tag{67}$$

Thus, the proposition is proved. $\square$

## D.3 Proof of Proposition 4.3

**Proposition D.4.** *For* $\mathbf{W} \in \mathbb{R}^{m \times n}$, *we denote* $|\mathbf{W}| = \mathbf{U}\boldsymbol{\Sigma}\mathbf{V}^{\top}$ *its* SVD. $\mathbf{s}_{\text{in}}$ *and* $\mathbf{s}_{\text{out}}$ *are given as:* $\mathbf{s}_{\text{in}} = \sqrt{\sigma_1}\mathbf{V}_{[:,1]}$, *and* $\mathbf{s}_{\text{out}} = \sqrt{\sigma_1}\mathbf{U}_{[:,1]}$. *We decompose the matrix as* $\mathbf{W} = \mathbf{W}_{\text{bool}} \odot |\mathbf{W}| \approx \mathbf{W}_{\text{bool}} \odot (\mathbf{s}_{\text{out}}\mathbf{s}_{\text{in}}^{\top})$. *We then have:*

$$\left\|\mathbf{W} - \mathbf{W}_{\text{bool}} \odot \mathbf{s}_{\text{out}}\mathbf{s}_{\text{in}}^{\top}\right\|_F^2 \leq \left\|\mathbf{W} - \mathbf{W}_{\text{bool}} \odot \mathbf{c}\mathbf{d}^{\top}\right\|_F^2, \quad \forall \mathbf{c} \in \mathbb{R}^{m \times 1}, \forall \mathbf{d} \in \mathbb{R}^{n \times 1}. \tag{68}$$

*Proof.* Similar to the proof of Proposition 4.3, we denote the following error matrices $\mathbf{E}_1 = |\mathbf{W}| - \mathbf{s}_{\text{out}}\mathbf{s}_{\text{in}}^{\top}$ and $\mathbf{E}_2 = |\mathbf{W}| - \mathbf{c}\mathbf{d}^{\top}$. We have that

$$\mathbf{W}_{\text{bool}} \odot |\mathbf{W}| - \mathbf{W}_{\text{bool}} \odot \mathbf{s}_{\text{out}}\mathbf{s}_{\text{in}}^{\top} = \mathbf{W}_{\text{bool}} \odot \mathbf{E}_1 \tag{69}$$

$$\Leftrightarrow \mathbf{W} - \mathbf{W}_{\text{bool}} \odot \mathbf{s}_{\text{out}}\mathbf{s}_{\text{in}}^{\top} = \mathbf{W}_{\text{bool}} \odot \mathbf{E}_1. \tag{70}$$

Therefore,

$$\left\|\mathbf{W} - \mathbf{W}_{\text{bool}} \odot \mathbf{s}_{\text{out}}\mathbf{s}_{\text{in}}^{\top}\right\|_F^2 = \|\mathbf{W}_{\text{bool}} \odot \mathbf{E}_1\|_F^2 = \sum_{i,j} \mathbf{W}_{\text{bool}[i,j]}^2 \mathbf{E}_{1[i,j]}^2 = \sum_{i,j} \mathbf{E}_{1[i,j]}^2 = \|\mathbf{E}_1\|_F^2. \tag{71}$$

Similarly, we have that

$$\left\|\mathbf{W} - \mathbf{W}_{\text{bool}} \odot \mathbf{a}\mathbf{b}^{\top}\right\|_F^2 = \|\mathbf{E}_2\|_F^2. \tag{72}$$

Thus, we need to show that

$$\|\mathbf{E}_1\|_F^2 \leq \|\mathbf{E}_2\|_F^2 \tag{73}$$

Additionally, we denote the rank-$k$ approximation to $|\mathbf{W}|$ by SVD as $\mathbf{S}_k$:

$$\mathbf{S}_k = \sum_{i=1}^{k} \sigma_i \mathbf{U}_{[:,i]} \mathbf{V}_{[:,i]}^{\top}. \tag{74}$$

With this notation, we have that $\mathbf{S}_1 = \mathbf{s}_{\text{out}} \mathbf{s}_{\text{in}}^{\top}$ is the rank-1 approximation of $|\mathbf{W}|$ by SVD.

From Eq. 73, we need to show that if there is an arbitrary rank-1 approximation to $|\mathbf{W}|$, $\mathbf{P}_1 = \mathbf{c}\mathbf{d}^{\top}$, we then have

$$\left\| |\mathbf{W}| - \mathbf{s}_{\text{out}} \mathbf{s}_{\text{in}}^{\top} \right\|_F^2 \leq \left\| |\mathbf{W}| - \mathbf{c}\mathbf{d}^{\top} \right\|_F^2. \tag{75}$$

This can be done by using the Eckart-Young-Mirsky theorem (Eckart & Young, 1936). First, we have that

$$\left\| |\mathbf{W}| - \mathbf{S}_1 \right\|_F^2 = \left\| |\mathbf{W}| - \mathbf{s}_{\text{out}} \mathbf{s}_{\text{in}}^{\top} \right\|_F^2 = \left\| \sum_{i=2}^{n} \sigma_i \mathbf{U}_{[:,i]} \mathbf{V}_{[:,i]}^{\top} \right\|_F^2 = \sum_{i=2}^{n} \sigma_i^2. \tag{76}$$

By the triangle inequality with the spectral norm, if $|\mathbf{W}| = \mathbf{C} + \mathbf{D}$ then $\sigma_1(|\mathbf{W}|) \leq \sigma_1(\mathbf{C}) + \sigma_1(\mathbf{D})$. Suppose the $\mathbf{C}_k$ and $\mathbf{D}_k$ denote the rank-$k$ approximation to $\mathbf{C}$ and $\mathbf{D}$ by SVD method, respectively. Then, for any $i, j \geq 1$ we have

$$\sigma_i(\mathbf{C}) + \sigma_j(\mathbf{D}) = \sigma_1(\mathbf{C} - \mathbf{C}_{i-1}) + \sigma_1(\mathbf{D} - \mathbf{D}_{j-1}) \tag{77}$$
$$\geq \sigma_1(|\mathbf{W}| - \mathbf{C}_{i-1} - \mathbf{D}_{j-1}) \tag{78}$$
$$\geq \sigma_1(|\mathbf{W}| - \mathbf{S}_{i+j-2}) \quad (\text{since rank}(\mathbf{C}_{i-1} + \mathbf{D}_{j-1}) \leq i + j - 2) \tag{79}$$
$$= \sigma_{i+j-1}(|\mathbf{W}|). \tag{80}$$

Because $\sigma_2(\mathbf{P}_1) = 0$, when $\mathbf{C} = |\mathbf{W}| - \mathbf{P}_1$ and $\mathbf{D} = \mathbf{P}_1$ we have that for $i \geq 1, j = 2$, $\sigma_i(|\mathbf{W}| - \mathbf{P}_1) \geq \sigma_{i+1}(|\mathbf{W}|)$. As a result,

$$\| |\mathbf{W}| - \mathbf{P}_1 \|_F^2 = \sum i = 1^n \sigma_i(|\mathbf{W}| - \mathbf{P}_1)^2 \geq \sum i = 2^n \sigma_i(|\mathbf{W}|)^2 = \| |\mathbf{W}| - \mathbf{S}_1 \|_F^2 \tag{81}$$
$$\Leftrightarrow \|\mathbf{E}_2\|_F^2 \geq \|\mathbf{E}_1\|_F^2 \tag{82}$$
$$\Leftrightarrow \left\| \mathbf{W} - \mathbf{W}_{\text{bool}} \odot \mathbf{c}\mathbf{d}^{\top} \right\|_F^2 \geq \left\| \mathbf{W} - \mathbf{W}_{\text{bool}} \odot \mathbf{s}_{\text{out}} \mathbf{s}_{\text{in}}^{\top} \right\|_F^2. \tag{83}$$

Hence the proposition is proved. $\qquad\square$

# E  DETAILS ON KERNEL ALLOCATION

## E.1  WEIGHT IMPORTANCE ESTIMATION

We assess the importance of a linear weight in the original FP model by comparing the representations at its input and output. Let $\mathbf{X} \in \mathbb{R}^{d \times n}$ and $\mathbf{Y} \in \mathbb{R}^{d \times m}$ denote the input and output matrices of a linear layer, respectively, where $d$ is the number of samples, and $n$ and $m$ are the input and output feature dimensions. We hypothesize that a weight is important if it significantly transforms the input representations. For example, a weight matrix equivalent to the identity does not alter the representations and thus would be considered unimportant. To quantify this transformation, we use a robust metric for comparing neural representations.

Various similarity measures can be used for this purpose, such as cosine similarity, as done in (Gromov et al., 2025). In this work, we adopt PWCCA Morcos et al. (2018), which is particularly well-suited for our setting: it is invariant to linear transformations—an essential property given that large language models (LLMs) are primarily composed of linear layers—and effectively captures shared structure while filtering out noise Morcos et al. (2018).

Specifically, we define the importance score as:

$$h = 1 - \frac{1}{c} \sum_{i=1}^{c} \rho_{\text{PWCCA},i}(\mathbf{X}, \mathbf{Y}), \tag{84}$$

where $c$ denotes the number of canonical vectors used in the comparison (typically, $c = \min(n, m)$). The matrices $\mathbf{X}$ and $\mathbf{Y}$ are obtained by simply forwarding a set of data samples through the network. In our experiments, we use 128 random samples from the WikiText2 training set to estimate the importance score. Here, $\rho_{\text{PWCCA},i}$ represents the projection-weighted correlation along the $i$-th canonical direction. The following section describes in detail how this correlation is computed.

---

**Algorithm 9:** Kernel allocation.

```
1  Input
2  │   T ≥ 1 ;                                      /* model expansion limit */
3  │   E = [e_l^[k]] ∈ R^(N_W × K_max) for k ∈ [1, K_max], l ∈ [1, N_W] ;  /* residual approx error */
4  │   h = [h_l] ∈ R^(N_W × 1) ;                    /* weight importance scores */
5  │   p = [p_l] ∈ R^(N_W × 1) ;                    /* weight size ratios */
6  Initialize
7  │   k = [1, ..., 1]^T of length N_W ;            /* starting choice */
8  │   f = k < K_max ;                              /* feasible indicator */
9  │   C = (1/p log 1/p) ⊙ h ⊙ E ;       /* where ⊙ is broadcasted over E columns */
10 While not all f is False do
11 │   g := ∅, l := ∅;
12 │   for l = 1 : N_W do
13 │   │   if f[l] = True then
14 │   │   │   g := C[l, k[l]] − C[l, k[l] + 1] ;    /* gain by increasing kernel size by 1 */
15 │   │   │   Append l to l, append g to g;
16 │   Sort g in decreasing order, and arrange l accordingly;
17 │   for (g, l) in (g, l) do
18 │   │   k_l := k;
19 │   │   k_l[l] = k_l[l] + 1;
20 │   │   if k_l^T p ≤ T then
21 │   │   │   k[l] = k[l] + 1;
22 │   │   │   break ;                              /* escape the for loop */
23 │   │   else
24 │   │   │   f[l] := False;
25 │   f ← and(f, k < K_max) ;                      /* element-wise logical and */
26 return k
```

---

**Projection-weighted Canonical Correlation Analysis.**  Canonical Correlation Analysis (CCA) finds bases for two matrices such that, when the original matrices are projected onto these bases, the

resulting projections are maximally correlated. Without loss of generality, we assume that $n \leq m$. For $1 \leq i \leq n$, the $i$-th canonical correlation coefficient $\rho_i$ is given by:

$$\rho_i = \max_{\mathbf{w}_{\mathbf{X}}^i, \mathbf{w}_{\mathbf{Y}}^i} \text{corr}(\mathbf{X}\mathbf{w}_{\mathbf{X}}^i, \mathbf{Y}\mathbf{Y}\mathbf{w}_{\mathbf{Y}}^i) \tag{85}$$

$$\text{subject to} \quad \mathbf{X}\mathbf{w}_{\mathbf{X}}^i \perp \mathbf{X}\mathbf{w}_{\mathbf{X}}^j \quad \forall j < i$$

$$\mathbf{Y}\mathbf{w}_{\mathbf{Y}}^i \perp \mathbf{Y}\mathbf{w}_{\mathbf{Y}}^j \quad \forall j < i.$$

The vectors $\mathbf{w}_{\mathbf{X}}^i \in \mathbb{R}^n$ and $\mathbf{w}_{\mathbf{Y}}^i \in \mathbb{R}^m$ that maximize $\rho_i$ are called the canonical weights. These weights transform the original data into the canonical variables $\mathbf{X}\mathbf{w}_{\mathbf{X}}^i$ and $\mathbf{Y}\mathbf{w}_{\mathbf{Y}}^i$. The constraints in Eq. 85 enforce orthogonality among the canonical variables, ensuring that each successive pair captures a distinct mode of correlation.

The mean CCA correlation is then computed as:

$$\bar{\rho}_{\text{CCA}} = \frac{\sum_{i=1}^n \rho_i}{n}, \tag{86}$$

where $n$ is the number of canonical correlation coefficients considered.

CCA is sensitive to perturbation when the condition number of $\mathbf{X}$ and $\mathbf{Y}$ is large. To imporve robustness, Morcos et al. (2018) propose a strategy to reduce this sensitivity, which they term "projection-weighted CCA" (PWCCA).

$$\rho_{\text{PWCCA},i} = \frac{\sum_{i=1}^c \alpha_i \rho_i}{\sum_{i=1}^c \alpha_i}, \quad \alpha_i = \sum_j |\langle \mathbf{h}_i, \mathbf{x}_j \rangle|, \tag{87}$$

where $\mathbf{x}_j$ is the $j$-th column of $\mathbf{X}$, and $\mathbf{h}_i = \mathbf{X}\mathbf{w}_{\mathbf{X}}^i$ is the vector of canonical variables formed by projecting $\mathbf{X}$ to the $i$-th canonical cooridate frame.

### E.2 KERNEL ALLOCATION ALGORITHM

Algorithm 9 illustrates the details of our algorithm for kernel allocation.

## F THEORETICAL ANALYSIS OF TRAINING COMPLEXITY

Consider a linear layer without bias, defined as $\mathbf{Y} = \mathbf{X}\mathbf{W}$ where $\mathbf{X} \in \mathbb{R}^{B \times L \times N}$ and $\mathbf{W} \in \mathbb{R}^{N \times M}$. Here, $B$ is the mini-batch size, $L$ is the sequence length, $N$ is the input dimension, and $M$ is the output dimension. We analyze the number of multiplications (MULs) required.

**Latent-weight approach (same cost as full-precision training):**

- Forward: $B \times L \times N \times M$ (FP16–FP16 MULs)
- Backward w.r.t. weights: $B \times L \times N \times M$ (FP16–FP16 MULs)
- Backward w.r.t. inputs: $B \times L \times N \times M$ (FP16–FP16 MULs)
- **Total:** $3 \times B \times L \times N \times M$ FP16–FP16 MULs

**Boolean approach with $K$ kernels:** (assuming FP16 gradients for a fair comparison). As shown in the main text, only the final Boolean kernel needs to be fine-tuned. The number of multiplications becomes:

- Forward: $K \times B \times L \times N \times M$ (BOOL–FP16 MULs, using all $K$ kernels)
- Backward w.r.t. weights: $1 \times B \times L \times N \times M$ (FP16–FP16 MULs, for last kernel only)
- Backward w.r.t. inputs: $1 \times B \times L \times N \times M$ (BOOL–FP16 MULs, for last kernel only)
- **Total:** $(K+1) \times B \times L \times N \times M$ BOOL–FP16 MULs, and $B \times L \times N \times M$ FP16–FP16 MULs

Since $K$ is typically small (e.g., 2–4) while $B$ and $L$ are large (thousands), most computation shifts from FP16–FP16 to the more efficient BOOL–FP16 operations. If we ignore the BOOL–FP16 MULs,

the FP16–FP16 operations are reduced by a factor of $2/3$ (i.e., a 66.7% reduction). Remarkably, this reduction is achieved while using more kernels and attaining better performance, yet with significantly lower training complexity. According to BitNet (Wang et al., 2023) (Table 1), for $L = 512$ and a LLaMA-like 13B model on 7 nm hardware, 1Bit–FP16 operations yield an energy saving of approximately **56×** compared to FP16–FP16. Hence, our method achieves substantial training efficiency. Importantly, BitNet is a latent-weight approach, with efficiency gains realized primarily during inference, whereas our method provides significant benefits already during training and fine-tuning.

We note that the above analysis does not include optimizer cost. The latent-weight approach typically relies on Adam, which requires two full-precision momenta per parameter and a complex update rule involving multiple normalization statistics. By contrast, our Boolean approach employs a Boolean optimizer requiring only one full-precision momentum per parameter, coupled with a much simpler update rule (see Eq. 3). This further underscores the reduction in overall training complexity offered by our method.

# G    ADDITIONAL EXPERIEMENTAL RESULTS

## G.1    ADDITIONAL INFORMATION OF EXPERIEMENTAL SETTINGS

We use 12 Nvidia GPUs of Tesla V100 for our experiments. We follow exactly the experimental settings in Jo et al. (2024). The results of the baselines in Table 2 are taken from Xu et al. (2024); Jo et al. (2024).

## G.2    ON THE CHOICE OF KD LOSS

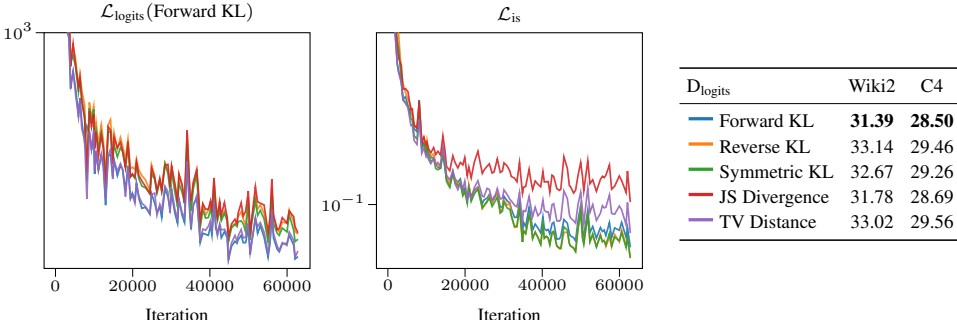

Figure 12: The training convergence of $\mathcal{L}_{\text{is}}$, and $\mathcal{L}_{\text{logits}}$, measured by Forward KL, and the final results with respect to the choice of $\text{D}_{\text{logits}}$.

Fig. 12 illustrates the convergence and results of using different choices for $\text{D}_{\text{logits}}$ in Eq. 10. Despite its simplicity, forward KL achieves the best performance. More complex measures, such as total variance (TV) distance (Wen et al., 2023) and Jensen-Shannon (JS) divergence (Agarwal et al., 2024), offer no significant benefits in our case. Furthermore, we observe that the final perplexity is strongly correlated with $\mathcal{L}_{\text{logits}}$ using forward KL, but not with $\mathcal{L}_{\text{is}}$, as shown in Fig. 12 and Fig. 6. As a result, we employ the forward KL in all experiments.

## G.3    RESULTS OF DIFFERENT NUMBER OF KERNELS ON LLMS

To complement the Table 2, Table 6 shows the benchmarking results of LLMs using our MBOK method with varying numbers of kernels per weight. Consistent with the observations made on smaller models in § 6.1.1, we observe that increasing the number of kernels generally improves performance. However, the performance gains begin to diminish noticeably beyond three kernels.

Table 6: Perplexity and zero-shot accuracy results of our MBOK method with different number of kernels.

| Model | Method | Wbits | Perplexity (↓) | | Zero-shot Accuracy (↑) | | | | | | |
|---|---|---|---|---|---|---|---|---|---|---|---|
| | | | Wiki2 | C4 | BoolQ | PIQA | Hella. | WinoG. | ARC-e | ARC-c | Average |
| OPT-1.3B | MBOK (2 kernels) | 2×1 | 16.13 | 16.61 | 58.53 | 70.67 | 48.11 | 56.75 | 48.19 | 27.90 | 51.69 |
| | MBOK (3 kernels) | 3×1 | 15.30 | 15.68 | 60.64 | 70.78 | 50.71 | 56.83 | 48.82 | 28.49 | 52.71 |
| | MBOK (4 kernels) | 4×1 | **14.83** | **14.92** | 60.95 | 70.85 | 51.02 | 56.85 | 49.13 | 29.24 | **53.01** |
| LLaMA-7B | MBOK (2 kernels) | 2×1 | 6.83 | 8.53 | 69.20 | 74.32 | 64.80 | 60.30 | 49.05 | 34.90 | 58.76 |
| | MBOK (3 kernels) | 3×1 | 6.20 | 7.76 | 67.89 | 76.15 | 68.91 | 63.30 | 48.94 | 37.62 | 60.47 |
| | MBOK (4 kernels) | 4×1 | **6.01** | **7.53** | 68.16 | 76.71 | 69.85 | 62.09 | 49.24 | 38.14 | **60.70** |
| LLaMA-13B | MBOK (2 kernels) | 2×1 | 6.17 | 7.88 | 68.10 | 76.33 | 69.88 | 64.17 | 52.34 | 37.88 | 61.45 |
| | MBOK (3 kernels) | 3×1 | 5.58 | 7.15 | 67.39 | 77.74 | 73.37 | 66.61 | 54.04 | 41.21 | 63.39 |
| | MBOK (4 kernels) | 4×1 | **5.38** | **6.91** | 68.69 | 77.63 | 74.23 | 66.53 | 56.14 | 41.38 | **64.10** |

## G.4 ADDITIONAL RESULTS ON LLaMA-2

Table 7 shows the results on LLaMA2-13B (Touvron et al., 2023b). Similar to the Table 2, the results of the baselines are taken from Xu et al. (2024) and Jo et al. (2024). It is clear that our method consistently outperforms the baselines across different metrics and model sizes. This further emphasizes the robustness of our approach across various types of models.

Table 7: Perplexity and zero-shot accuracy results of Float16, quantized and binarized LLaMA2 models.

| Model | Method | Wbits | Perplexity (↓) | | Zero-shot Accuracy (↑) | | | | | | |
|---|---|---|---|---|---|---|---|---|---|---|---|
| | | | Wiki2 | C4 | BoolQ | PIQA | Hella. | WinoG. | ARC-e | ARC-c | Average |
| LLaMA2-7B | FP16 | 16 | 5.47 | 6.97 | 71.10 | 76.88 | 72.94 | 67.09 | 53.58 | 40.61 | 63.70 |
| | PB-LLM | 1.7 | 76.75 | 85.92 | 62.17 | 52.82 | 26.87 | 50.11 | 26.89 | 24.31 | 40.53 |
| | BiLLM | 1.11 | 27.72 | 36.34 | 62.14 | 59.19 | 35.18 | 53.11 | 34.22 | 26.54 | 45.06 |
| | OneBit | 1 | 8.60 | 10.74 | 63.06 | 70.40 | 54.24 | 56.67 | 40.82 | 29.35 | 52.42 |
| | MoS | 1 | 7.88 | 9.75 | 65.02 | 71.55 | 59.41 | 56.18 | 41.84 | 30.03 | 54.01 |
| | OPTQ | 2 | 7.7e3 | NaN | 42.97 | 49.46 | 26.19 | 50.28 | 26.77 | 28.58 | 37.38 |
| | LLM-QAT | 2 | 1.1e3 | 6.6e2 | 59.14 | 50.12 | 25.10 | 49.08 | 26.26 | 26.96 | 35.89 |
| | OmniQuant | 2 | 31.21 | 64.34 | 58.69 | 56.53 | 33.87 | 51.22 | 33.63 | 24.32 | 43.12 |
| | MBOK [Ours] | 2×1 | **6.87** | **8.74** | 66.94 | 74.97 | 65.59 | 61.72 | 44.82 | 34.21 | **58.04** |
| | MBOK [Ours] | 3×1 | **6.12** | **7.81** | 65.46 | 75.79 | 69.59 | 62.04 | 49.11 | 37.80 | **59.97** |
| LLaMA2-13B | FP16 | 16 | 4.88 | 6.47 | 68.99 | 79.05 | 76.62 | 69.77 | 57.95 | 44.20 | 66.10 |
| | PB-LLM | 1.7 | 155.25 | 151.15 | 37.82 | 53.26 | 28.89 | 49.48 | 28.28 | 23.72 | 36.91 |
| | BiLLM | 1.11 | 20.71 | 27.19 | 62.20 | 62.51 | 38.05 | 56.35 | 40.69 | 27.73 | 47.92 |
| | OneBit | 1 | 7.56 | 9.67 | 65.66 | 71.60 | 60.07 | 56.91 | 45.76 | 31.74 | 55.29 |
| | MoS | 1 | 7.08 | 8.91 | 66.12 | 73.72 | 63.80 | 58.98 | 45.71 | 33.19 | 57.09 |
| | OPTQ | 2 | 2.1e3 | 3.2e2 | 40.61 | 51.74 | 25.67 | 51.85 | 25.46 | 27.30 | 37.11 |
| | LLM-QAT | 2 | 5.1e2 | 1.1e3 | 39.85 | 49.08 | 24.37 | 51.38 | 27.15 | 24.32 | 36.03 |
| | OmniQuant | 2 | 16.88 | 27.02 | 62.05 | 62.24 | 50.34 | 53.20 | 40.66 | 29.61 | 49.68 |
| | MBOK [Ours] | 2×1 | **5.97** | **7.85** | 66.32 | 75.84 | 70.24 | 62.51 | 50.00 | 37.46 | **60.40** |
| | MBOK [Ours] | 3×1 | **5.35** | **7.07** | 66.80 | 77.59 | 73.79 | 65.27 | 54.47 | 41.89 | **63.30** |

## G.5 GENERATION QUALITY

Table 8 illustrates the generated text given zero-shot arbitrary questions from LLaMA-13B with FP16 weights and MBOK using three kernels. It is clearly that our model is able to produce meaningful and coherennt text compared to the original FP16 model.

Table 8: Comparison of generation quality (*non-curated*) on LLaMA-13B models with FP16 and MBOK using three kernels.

| | |
|---|---|
| **Prompt:** | What are the capitals of China, US and Russia? |
| **FP16:** | The capital of China is Beijing. The capital of the United States is Washington D.C. and the capital of Russia is Moscow. |
| **MBOK:** | The capital of China is Beijing. The capital of the US is Washington D.C. and the capital of Russia is Moscow. |
| **Prompt:** | I have a friend who lives in London and I would like to visit him. How would I go there from Paris? |
| **FP16:** | You can take the Eurostar from Gare du Nord in Paris to St. Pancras Station in London. The trip takes about 2 hours and 15 minutes. |
| **MBOK:** | The fastest way to get from Paris to London is to take the Eurostar train. The Eurostar departs from the Gare du Nord station in Paris and arrives at the St. Pancras International station in London. The trip takes about 2 hours and 15 minutes. |
| **Prompt:** | What is Fete du Travail? |
| **FP16:** | It is an annual holiday celebrated all over the world that resulted from the labour union movement, to celebrate the economic and social achievements of workers. The majority of countries celebrate Labour Day on May Day, or May 1, and it is popularly known as May Day and International Workers' Day |
| **MBOK:** | In France, May 1st is a public holiday. It is also known as Labour Day, International Workers_ Day, or May Day. It is a celebration of the social and economic achievements of the labour movement. It originated from the eight-hour day movement, which advocated eight hours for work, eight for recreation, and eight for rest. |
| **Prompt:** | What are the main ingredients in pho, Vietnam_s iconic noodle soup? |
| **FP16:** | Rice noodles, beef or chicken, onions, ginger, cilantro, bean sprouts, basil, lime, hoisin sauce, Sriracha, and fish sauce. |
| **MBOK:** | Bone broth, rice noodles, fish sauce, herbs and spices, and beef, chicken, or seafood, depending on the type of pho you_re eating. Pho is usually served with bean sprouts, Thai basil, lime wedges, and hoisin and sriracha sauces. |

## G.6 ADDITIONAL BASELINES

### G.6.1 COMPARISONS WITH QUIP AND SHIFTADDLLM

Both QuIP (Chee et al., 2023) and ShiftAddLLM (You et al., 2024) are PTQ method for LLMs. QuIP is a two-step process that leverages the insight that quantization performs better when weight and Hessian matrices are incoherent. It uses an adaptive rounding procedure to minimize a quadratic proxy objective, which measures the error between the original and quantized weights. Additionally, it applies pre- and post-processing steps using random orthogonal matrices to ensure the weight and Hessian matrices are incoherent. Conversely, our method does not employ either these complicated pre- and post-processing steps or costly Hessian matrices. Meanwhile, ShiftAddLLM is a post-training

reparameterization process, which quantizes each weight matrix in the LLM into a set of binary matrices and group-wise scaling factors. he original multiplication between activations and weights is then reparameterized into: (1) bitwise shifts for the activations, using the power-of-two quantized scaling factors, and (2) additions of the results, guided by the binary weight matrices; this process can be implemented using look-up tables (LUTs) on GPUs.

Table 9 presents results on OPT models, with competitor results extracted from their respective original papers. Notably, ShiftAddLLM utilizes a more computationally expensive group quantization, whereas our method does not. Our results clearly demonstrate that our approach consistently and significantly outperforms these baselines, particularly in the 2-bit scenario.

Table 9: Comparisons with QuIP, ShiftAddLLM using OPT models.

| BIT-WIDTH | METHOD | OPT-125M | OPT-350M | OPT-1.3B |
|---|---|---|---|---|
| 2 | QuIP (Chee et al., 2023) | 34.22 | 25.19 | 16.21 |
| | ShiftAddLLM (You et al., 2024) | 31.29 | 24.24 | 21.53 |
| | MBOK [Ours] | **29.10** | **23.12** | **15.03** |
| 3 | QuIP (Chee et al., 2023) | 347.40 | 672.30 | 41.64 |
| | ShiftAddLLM (You et al., 2024) | 51.15 | 40.24 | 29.03 |
| | MBOK [Our] | **28.60** | **24.54** | **16.13** |

### G.6.2 COMPARIONS WITH BITSTACK, DB-LLM AND AWQ

While BitStack (Wang et al., 2025) also decompose weights using SVD, its core method and goal fundamentally differ from our method. BitStack is a training-free method primarily aimed at saving storage for inference. In contrast, our method not only converts FP models into Boolean models but also includes further fine-tuning, with the goal of achieving low complexity in both training and inference. Furthermore, while BitStack packs the extracted binary matrix into GPU-supported data types to reduce inference memory, and its approach to loading residual blocks relies on their influence on perplexity, our approach to residual block management is distinct.

DB-LLM (Chen et al., 2024) is limited to a fixed decomposition into two binary matrices, whereas our MBOK method generalizes to an arbitrary number of Boolean kernels. In DB-LLM, the full-precision knowledge is preserved only through scaling factors and binary matrices derived implicitly via thresholding. There is no formal analysis proving the optimality of this formulation. In contrast, thanks to the SVID in our approach, each extracted kernel is accompanied by an optimal scaling vector and Boolean matrix. This allows us to only finetune the last kernel to calibrate the entire model. Like most existing binary LLMs, DB-LLM relies on full-precision latent weights during training and finetuning. Our method does not require this, as it directly operates in the Boolean domain. This distinction is particularly important in the LLM context, where training and finetuning can be computationally expensive.

Table 10 compares our method, MBOK (with 2 kernels), against BitStack, DB-LLM, and AWQ (Lin et al., 2024) on LLaMA2-7B. It is evident that our method consistently outperforms all baselines.

Table 10: Comparisons with AWQ, BitStack, DB-LLM using LLaMA2-7B with 2-bit setting.

| METHOD | Wiki2 ($\downarrow$) | ARc-e ($\uparrow$) | ARC-c ($\uparrow$) | PIQA ($\uparrow$) | Hella. ($\uparrow$) | WinoG. ($\uparrow$) |
|---|---|---|---|---|---|---|
| AWQ (Lin et al., 2024) | 1.8e5 | 26.3 | 26.7 | 50.9 | 26.5 | 49.3 |
| BitStack (Wang et al., 2025) | 29.93 | 32.3 | 25.6 | 62.4 | 42.8 | 53.6 |
| DB-LLM (Chen et al., 2024) | 7.23 | **45.2** | 33.5 | 73.1 | 61.9 | 61.7 |
| MBOK [Ours] | **6.87** | 44.8 | **34.2** | **75.0** | **65.6** | **61.7** |

## G.7 EFFECTS OF KNOWLEDGE DISTILLATION

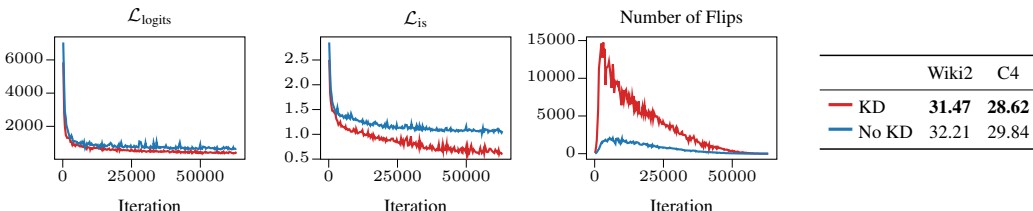

Figure 13: Study on the effect of using knowledge distillation on OPT-125M with 2 Boolean kernels.

Fig. 13 presents a comparison between training with and without Knowledge Distillation (KD). It is evident that employing KD outperforms the baseline in terms of test perplexity on the Wiki2 and C4 datasets, as it provides more informative guidance during training. To investigate this behavior further, we visualize the convergence of $\mathcal{L}_{\text{logits}}$ and $\mathcal{L}_{\text{is}}$. Aided by the informative guidance from the teacher, convergence with KD is significantly faster. Furthermore, the model learns more effectively—leveraging the additional signal from the teacher—as evidenced by the higher flipping rates compared to training without KD.

## G.8 ANALYSIS OF SCALING VALUES

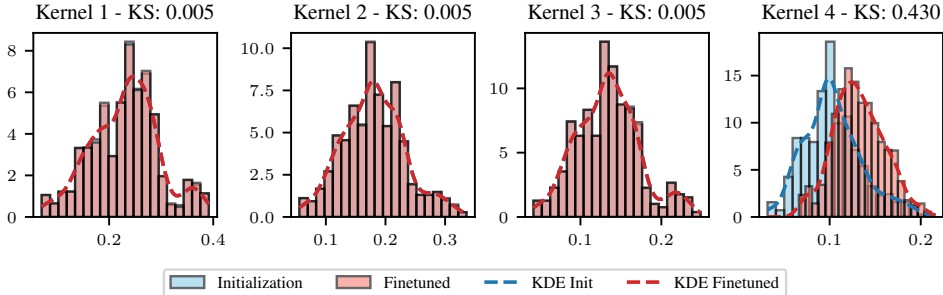

Figure 14: Histogram of output-scaling values for the first linear layer of OPT-125M with four kernels, shown at initialization and after finetuning. The Kolmogorov–Smirnov (KS) statistic is also reported to quantify the difference between the scaling-value distributions before and after finetuning.

Fig. 14 illustrates how the output-scaling values of the four kernels in the first linear layer change from initialization to after finetuning. All output-scaling values are learnable; however, only those associated with the last kernel exhibit a substantial shift during training. This is evident from both the histogram changes and the corresponding Kolmogorov–Smirnov distance.

After finetuning, the scaling values of the last kernel become significantly larger and more dominant, whereas the scaling values of the other kernels change only minimally. This observation supports our theoretical analysis: successive SVID extraction provides sufficiently strong initialization for the low-order kernels, and finetuning primarily the last kernel is already adequate.

## G.9 CONVERGENCES OF OPT MODELS

Fig. 15 shows the training convergences of MBOK using 3 kernels with OPT models.

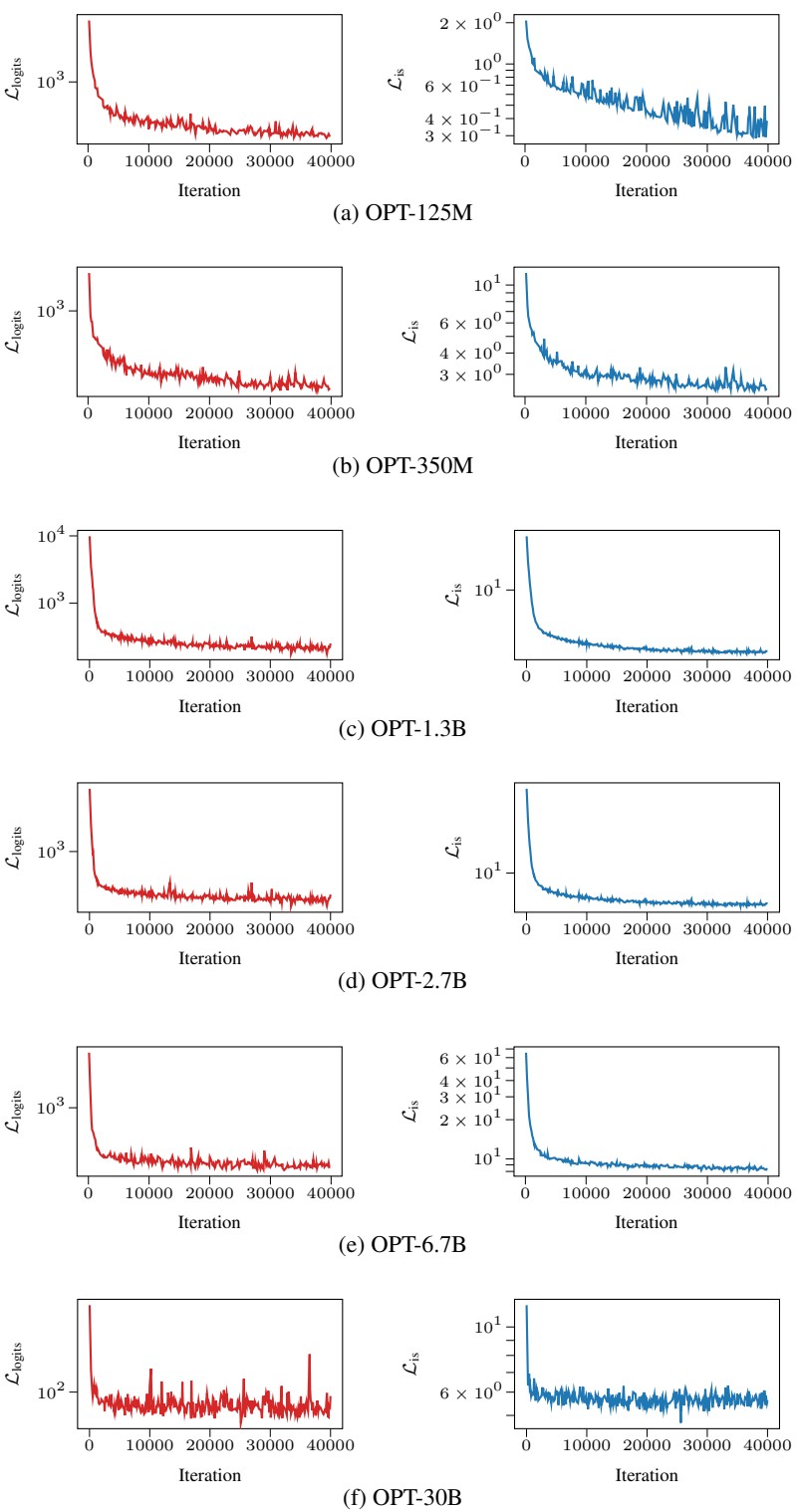

Figure 15: The training convergences of MBOK using 3 kernels with OPT models.

## G.10 EFFECTS OF SUCCESSIVE SVID INITIALIZATION

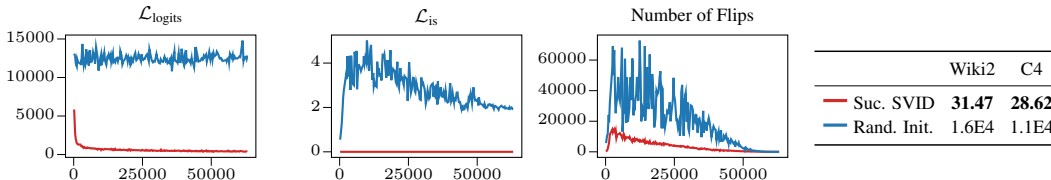

Figure 16: Study on the effect of using our successive SVID strategy and random initialization on OPT-125M with 2 Boolean kernels.

Fig. 16 compares our proposed successive SVID initialization with a random initialization. It is clear that our method delivers far better results, while the random initialization often fails to converge. Moreover, our initialization enables the model to learn efficiently, whereas the random initialization causes the model to struggle, as reflected by the large number of Boolean flips.

## G.11 DISCUSSION ON LATENCY AND COMPARISON WITH VECTOR QUANTIZATION

**Scalar and Vector Quantization.** In the context of LLMs, scalar quantization and vector quantization are two different approaches for compressing weights. Scalar quantization maps each weight or activation independently to a smaller set of discrete levels (e.g., 32-bit floating-point to 8- or 4-bit integers). It is simple, hardware-friendly, and widely used in practice, but it ignores correlations across dimensions, potentially discarding fine-grained structure. Vector quantization (VQ) instead compresses entire vectors (e.g., weight groups) by replacing them with indices into a learned codebook of representative vectors. By capturing cross-dimensional correlations, VQ often achieves higher compression, particularly for large embedding tables. However, codebook training is more complex, and inference requires index lookups to reconstruct vectors. This adds significant overhead to both quantization and dequantization, leading to much higher latency compared to scalar methods.

Our method is native 1-bit weight design, its nearest baselines are scalar weight quantization. As a result, for a fair comparison, in the main text we mainly consider state-of-the-art scalar quantization like OmniQuant (Shao et al., 2024), OPTQ (Frantar et al., 2023), LLM-QAT (Liu et al., 2024c) as the main baselines. Nevertheless, for completeness, we also compare our approach against state-of-the-art ultra low-bit vector quantization (VQ) methods for LLMs, including QTIP (Tseng et al., 2024b) and QUIP# (Tseng et al., 2024a) in a 2-bit setting, specifically on LLaMA-7B with a sequence length of 2048 (results taken from the QTIP paper). The results are summarized in Table 11. Remarkably, our method's performance is comparable to these state-of-the-art (SOTA) VQ methods. This is noteworthy given that our approach directly utilizes native Boolean weights, eliminating the need for the very costly quantization and dequantization of high-dimensional vectors inherent in VQ.

Table 11: Perplexity comparison with SOTA vector quantization methods using LLaMA-7B.

| METHOD | Wiki2 ($\downarrow$) | C4 ($\downarrow$) |
|---|---|---|
| QUIP# (Tseng et al., 2024a) | 6.86 | 8.36 |
| QTIP (Tseng et al., 2024b) | 6.52 | 7.99 |
| MBOK [Ours] | 6.83 | 8.53 |

**Empirical Evidence of Latency Gains.** To demonstrate the practicality of our approach even on modern hardware such as GPUs, we leverage the recently introduced BitBLAS library [1] (Wang et al., 2024) for 1-bit matrix multiplications. Using FP16 activations with INT1 weights, we measure the latency of linear layers in LLaMA-7B (Table 12) and LLaMA-13B (Table 13) under an inference batch size of 1, evaluating our method MBOK with two kernels. Our results show that MBOK achieves up to

---

[1]https://github.com/microsoft/BitBLAS

an $8.7\times$ speedup over FP16 baselines, while substantially outperforming existing binarization and scalar quantization methods, as detailed in the main text. We also benchmark against 2-bit QUIP# and QTIP using the authors' official implementations[23]. All experiments are conducted on an A100 GPU.

Remarkably, our method is not only much faster than these VQ baselines but also delivers comparable performance. This is expected, as VQ-based methods incur significant overhead from the costly encoding and decoding steps required to realize their high compression ratios. Taken together, the results highlight that our native Boolean approach offers a compelling and efficient alternative to state-of-the-art vector quantization methods. With dedicated Boolean hardware accelerators, the performance gains would be even more pronounced.

Table 12: Measured latency (ms) of linear layers in LLaMA-7B, with values in parentheses denoting speed-up relative to the FP16 baseline.

| WEIGHT SIZE | FP16 | QUIP# (Tseng et al., 2024a) | QTIP (Tseng et al., 2024b) | MBOK (Ours) |
|---|---|---|---|---|
| $4096 \times 4096$ | 0.10697 | 0.46196 ($0.23\times$) | 1.37137 ($0.08\times$) | **0.04989 ($2.14\times$)** |
| $4096 \times 11008$ | 0.27935 | 0.55526 ($0.50\times$) | 3.13633 ($0.09\times$) | **0.05136 ($5.44\times$)** |
| $11008 \times 4096$ | 0.27664 | 0.55988 ($0.49\times$) | 3.16067 ($0.09\times$) | **0.05117 ($5.41\times$)** |

Table 13: Measured latency (ms) of linear layers in LLaMA-13B, with values in parentheses denoting speed-up relative to the FP16 baseline.

| WEIGHT SIZE | FP16 | QUIP# (Tseng et al., 2024a) | QTIP (Tseng et al., 2024b) | MBOK (Ours) |
|---|---|---|---|---|
| $5120 \times 5120$ | 0.16540 | 0.62260 ($0.27\times$) | 1.96368 ($0.08\times$) | **0.05074 ($3.25\times$)** |
| $5120 \times 13824$ | 0.42830 | 0.62836 ($0.68\times$) | 5.23681 ($0.09\times$) | **0.05098 ($8.40\times$)** |
| $13824 \times 5120$ | 0.43411 | 0.62840 ($0.69\times$) | 5.21193 ($0.08\times$) | **0.04987 ($8.70\times$)** |

## H   ETHICS STATEMENT

This work makes a fundamental contribution to machine learning methodology. It does not involve human subjects, sensitive data, or applications with direct societal or ethical risks. We do not foresee any immediate ethical concerns arising from this research.

## I   REPRODUCIBILITY STATEMENT

We provide detailed descriptions of all algorithms and illustrative code for the core components. Experiments are conducted on standard benchmarks using established testing procedures, and all experimental details and settings are fully declared to facilitate independent reproduction of our results.

## J   THE USE OF LARGE LANGUAGE MODELS

We used large language models (LLMs) solely for non-substantive assistance, including grammar refinement and summarizing relevant literature. All research ideas, analyses, and conclusions are the authors' own.

---

[2]https://github.com/Cornell-RelaxML/quip-sharp
[3]https://github.com/Cornell-RelaxML/qtip

