# OpenReview forum: "Highly Efficient and Effective LLMs with Multi-Boolean Architectures"
_ICLR.cc/2026/Conference — ICLR 2026 Poster_

### Official Review · Reviewer_xHhd · 2025-10-24

**Soundness:** 3
**Presentation:** 3
**Contribution:** 2
**Rating:** 6
**Confidence:** 3

**Summary:**

This paper introduces MBOK, a binarization framework that trains LLMs directly in the Boolean domain with multiple Boolean kernels, avoiding latent weights. It achieves near-FP16 performance with lower memory and computation compared to existing binarization and quantization methods.

**Strengths:**

1. The paper addresses an important problem in LLM binarization and makes a convincing case for reducing the reliance on latent full-precision weights.
2. The proposed approach is technically sound and well-motivated, with clear formulations and ablation studies that support the design choices.
3. Experimental evaluation is fairly comprehensive, showing competitive or even near-FP16 performance with much lower memory and computation cost.
4. The presentation is clear, with tables and figures that make it easy to compare against strong baselines.

**Weaknesses:**

1. The paper claims Pareto-frontier results, but only compares against 2\3-bit quantization without presenting a full Pareto curve, which makes the claim less convincing.
2. Although the method is said to optimize directly in the Boolean domain, it is unclear whether training/finetuning actually reduces memory consumption, or if fake quantization is still used; concrete evidence of memory savings during optimization is missing.
3. The comparison with BinaryMoS seems unfair: three Boolean kernels likely consume more memory than three MoS experts, yet the paper does not account for this discrepancy.
4. No results are reported on real inference acceleration or end-to-end memory reduction; claims about efficiency remain theoretical.
5. Large-scale validation is missing — while OPT and LLaMA-13B are evaluated, experiments on larger models (e.g., 65B) are absent, leaving scalability uncertain.
6. The distinction between the proposed multi-Boolean kernels and existing multi-binary-base methods (e.g., BitStack, QBB, DB-LLM) is not sufficiently clarified, weakening the novelty claim.
7. The initialization strategy (SVID) is largely borrowed from OneBit, which reduces the originality of that component.

**Questions:**

Please refer to the weakness section.

---

> ### Author Response · Authors · 2025-11-21
> **Official Comment by Authors**
>
> Thanks reviewer for constructive feedback and for recognizing our method as "addresses an important problem", "convincing", "sound and well-motivated", "comprehensive". We appreciate the opportunity to clarify the confusing points and address the questions raised.
>
> ---
>
> **Q1. The paper claims Pareto-frontier results, but only compares against 2\3-bit quantization without presenting a full Pareto curve, which makes the claim less convincing.**
>
> ---
>
> Thank Reviewer for the feedback and the opportunity for us to further clarify on this point.
>
> Figure 1 and Table 3 plot Perplexity vs. Model Size (GB) across five different model scales (125M, 350M, 1.3B, 2.7B, 6.7B). As shown in Figure 1, the MBOK curve (green triangles) sits strictly below and to the left of the GPTQ curve (blue circles). This confirms that for any given memory budget (Model Size), MBOK yields lower perplexity than the baseline, thereby empirically establishing a superior Pareto frontier across the OPT model family.
>
> In addition, in our response to `Reviewer Mstc`, we have added newly advanced QAT methods for Table 3. Our method still outperforms these strong baselines. We believe that these results provides comprehensive evidence for our Pareto-frontier claim.
>
> ----
>
>
> **Q2. Q4. Although the method is said to optimize directly in the Boolean domain, it is unclear whether training/finetuning actually reduces memory consumption, or if fake quantization is still used; concrete evidence of memory savings during optimization is missing.  No results are reported on real inference acceleration or end-to-end memory reduction; claims about efficiency remain theoretical.**
>
> Thank you for your comment. As mentioned in the main paper, we rely on the BitBlas library, which enables efficient low-bit matrix multiplication with 1-bit weights. Below, we provide a benchmark on Google A100 instance. The results clearly demonstrate the advantages of our method.
>
> | Model | Method | End-to-end throughput | Speedup | Memory | Reduction |
> | :--- | :--- | :--- | :--- | :--- | :--- |
> | **LLaMa-2-7B** | FP-16 | 81.25 (tok/s) | 1x | 17.47 GB | 1x |
> | | MBOK (2 kernels) | 241.81 (tok/s) | 2.98x | 4.72 GB | 3.91x |
> | **LLaMa-2-13B** | FP-16 | 43.19 (tok/s) | 1x | 31.78 GB | 1x |
> | | MBOK (2 kernels) | 167.25 (tok/s) | 3.87x | 7.03 GB | 4.52x |
>
> ---
>
> **Q3. The comparison with BinaryMoS seems unfair: three Boolean kernels likely consume more memory than three MoS experts, yet the paper does not account for this discrepancy**
>
> Thank you for your constructive feedback.
> Indeed, as described in Section 6.4 of the initial version of the paper, we compared our method against latent-weight approaches using the same multi-Boolean structure. The only difference is that the baseline employs full-precision latent weights during training, which we chose to ensure the fairest possible comparison.
> Our results show that our method consistently outperforms this baseline.
>
> ---
>
> **Q5. Large-scale validation is missing — while OPT and LLaMA-13B are evaluated, experiments on larger models (e.g., 65B) are absent, leaving scalability uncertain.**
>
> Thank you for your comment. Unfortunenately, due to the limited computing resources during the rebuttal period, we could not conduct experiments on 70B models.
> However, we have just performed an experiment with  **OPT-30B model, a very large model**.
> The below table show the results of our method with 3 kernels compared to the 3-bit quantization baselines.
> As can be seen, our method can generalize very well from very small model (125M) to very large model (30B).
> Figure 15 in the Appendix, we show the training convergences on these models.
> These results  that our method exhibit training stability, convergence and scalibility across all considered model sizes.
> We hope these new results can address your concern.
>
> | Method | Wiki2 (125M) | Wiki2 (350M) | Wiki2 (1.3B) | Wiki2 (2.7B) | Wiki2 (6.7B) | Wiki2 (30B) | C4 (125M) | C4 (350M) | C4 (1.3B) | C4 (2.7B) | C4 (6.7B) | C4 (30B) |
> | :--- | :--- | :--- | :--- | :--- | :--- | :--- | :--- | :--- | :--- | :--- | :--- | :--- |
> | FP-16 | 27.65 | 22 | 14.63 | 12.47 | 10.86 | 9.56 | 26.56 | 22.59 | 16.07 | 14.34 | 12.71 | 11.44 |
> | RTN | 1.30E+03 | 64.57 | 1.30E+04 | 1.60E+04 | 5.80E+03 | 1.60E+03 | 834 | 55.49 | 5.20E+03 | 1.10E+04 | 5.30E+03 | 1.40E+03 |
> | OPTQ | 53.85 | 33.79 | 20.97 | 16.88 | 14.86 | 10.27 | 42.41 | 31.33 | 21.63 | 18.17 | 17.14 | 12.23 |
> | **MBOK** | **29.10** | **23.12** | **15.30** | **13.09** | **11.03** | **9.68** | **28.62** | **22.10** | **15.68** | **14.00** | **12.33** | **11.32** |

---

> > ### Author Response · Authors · 2025-11-21
> > **Official Comment by Authors**
> >
> > **Q6. The distinction between the proposed multi-Boolean kernels and existing multi-binary-base methods (e.g., BitStack, QBB, DB-LLM) is not sufficiently clarified, weakening the novelty claim**
> >
> > Thank you for your comment.
> > Indeed, due to the limited space in the main paper, we already compared carefully our method with BitStack and DB-LLM in the Appendix in the initial version.
> > In short, BitStack is a Training-Free (PTQ) method for storage. QBB/DB-LLM often rely on latent weights for training. MBOK's novelty is the Training-Aware (QAT) framework that optimizes multi-kernels directly in the Boolean domain without latent weights. This solves the "accuracy collapse" of PTQ while avoiding the "memory explosion" and "intensive computation" of standard QAT.
> >
> > ---
> >
> > **Q7. The initialization strategy (SVID) is largely borrowed from OneBit, which reduces the originality of that component.**
> >
> > Thank you for your comment.
> > In the paper we clearly explicitly credited OneBit for SVID in the initial version of the paper.
> > However, we additional provide further theoretical justification of SVID (Preposition 4.3) to show that it can extract optimal Boolean weights and scaling factors to motivate our successive SVID method.
> > In addition, OneBit uses SVID only for initialization before reverting to standard latent-weight training. Our contribution is the Successive SVID strategy combined with BOLD optimization, which allows us to discard the latent weights entirely—something OneBit cannot do.

---

> ### Comment · Reviewer_xHhd · 2025-11-25
>
> Thank you for the clarifications. I am satisfied with the responses and will maintain my score.

---

> > ### Author Response · Authors · 2025-11-28
> > **Official Response from Authors**
> >
> > Many thanks Reviewer for your time and consideration. We appreciate your thoughtful review and are glad the clarifications addressed your concerns. We are grateful that Reviewer maintains the positive opinion on our paper.

---

### Official Review · Reviewer_Mstc · 2025-10-28

**Soundness:** 2
**Presentation:** 3
**Contribution:** 2
**Rating:** 4
**Confidence:** 4

**Summary:**

This work proposes a multi-boolean-kernel QAT method for LLMs. Extending BOLD’s single-kernel design, it integrates successive SVID and adaptive mixed-precision, outperforming prior binarized QAT methods. It also enables direct boolean domain LLM fine-tuning to save training resources.

**Strengths:**

1. This work is supported by solid theoretical foundations, with a detailed explanation of the computational process of multi-boolean-kernels during training.
2. The approach of training only the last kernel is insightful, and the authors have provided detailed experimental comparisons and explanations to validate it.
3. Experimental results demonstrate that MBOK outperforms previous binarized QAT methods in both performance and resource efficiency.

**Weaknesses:**

1. The novelty of this work is limited, as its core contribution is merely a natural extension of BOLD, expanding from a single kernel to multiple kernels.
2. The authors claim that MBOK lies on the Pareto frontier. However, the evidence provided in Table 3 only compares it with low-bit PTQ methods such as RTN and GPTQ. Given that MBOK is a QAT method, a fairer evaluation would involve comparisons with advanced low-bit QAT methods like SpinQuant and EfficientQAT.
3. SpinQuant with W4A8 quantization can achieve better results (as shown in Table 1 of their paper) than MBOK with 4 kernels (i.e., W4A16). If compared under the same activation bit-width, SpinQuant may hold even greater advantages, which would undermine the scalability of multi-kernel designs (the core contribution of this work).
4. Key ablation studies are missing. Specifically, analyses are needed on how initialization via successive SVID impacts final distillation performance, and on the effectiveness of the kernel allocation strategy (i.e., mixed precision) compared to uniform precision under the same kernel budget. Including such ablation results would strengthen the work’s contribution and novelty.
5. Several typos and errors require correction. For instance, in Table 1, BiLLM also employs higher-bit salient weights. In Table 3, "QPTQ" should be corrected to "OPTQ".

**Questions:**

1. Is there any comparison of actual training time to demonstrate that MBOK is more efficient than other binarized QAT methods?
2. For other questions, please refer to the weaknesses.

I would be willing to reconsider my rating if the authors can address these concerns.

---

> ### Author Response · Authors · 2025-11-21
> **Official Comment by Authors**
>
> Thanks reviewer for constructive feedback and for recognizing our method as "supported by solid theoretical foundation", "insightful", "outperforms previous binarized QAT methods". We appreciate the opportunity to clarify the confusing points and address the questions raised. We believe that the new results will address the concerns and further strenthen the paper and thank the reviewer for triggering this.
>
> ---
>
> **Q1. However, the evidence provided in Table 3 only compares it with low-bit PTQ methods such as RTN and GPTQ. Given that MBOK is a QAT method, a fairer evaluation would involve comparisons with advanced low-bit QAT methods like SpinQuant and EfficientQAT.**
>
> We thank the reviewer for the constructive question.
>
> We conducted additional experiments with SpinQuant and EfficientQAT for Table 3 (OPT models with a 3-bit budget), using the author's official implementations. We observed that both SpinQuant and EfficientQAT are more computationally expensive than our method, as they require extra matrix multiplications for Hadamard-based rotations and group-wise quantization. In contrast, our approach introduces no additional matrix multiplications and remains closer in complexity to standard channel-wise quantization.
>
> For completeness and fairness, we report these baselines across all configurations. The table below presents the results. As shown, our method consistently outperforms both SpinQuant and EfficientQAT in all settings.
>
> | Method | Wiki2 (125M) | Wiki2 (350M) | Wiki2 (1.3B) | Wiki2 (2.7B) | Wiki2 (6.7B) | C4 (125M) | C4 (350M) | C4 (1.3B) | C4 (2.7B) | C4 (6.7B) |
> | :--- | :--- | :--- | :--- | :--- | :--- | :--- | :--- | :--- | :--- | :--- |
> | FP-16 | 27.65 | 22 | 14.63 | 12.47 | 10.86 | 26.56 | 22.59 | 16.07 | 14.34 | 12.71 |
> | SpinQuant (No Hadamard) | 34.02 | 26.92 | 16.12 | 13.98 | 12.13 | 30.98 | 28.53 | 18.24 | 15.84 | 14.73 |
> | SpinQuant (Hadamard) | 33.23 | 25.35 | 15.24 | 13.48 | 11.45 | 29.98 | 25.35 | 16.72 | 14.75 | 13.19 |
> | EfficientQAT (Channel) | 34.1 | 24.32 | 16.27 | 15.11 | 12.45 | 32.89 | 24.63 | 17.35 | 17.24 | 15.32 |
> | EfficientQAT (Group 128) | 31.03 | 24.08 | 15.71 | 13.42 | 11.56 | 29.12 | 24.39 | 16.89 | 15.12 | 13.32 |
> | **MBOK** | **29.1** | **23.12** | **15.3** | **13.09** | **11.03** | **28.62** | **22.1** | **15.68** | **14.00** | **12.33** |
>
> ---
>
> **Q2. SpinQuant with W4A8 quantization can achieve better results (as shown in Table 1 of their paper) than MBOK with 4 kernels (i.e., W4A16). If compared under the same activation bit-width, SpinQuant may hold even greater advantages, which would undermine the scalability of multi-kernel designs (the core contribution of this work).**
>
> Thank you for your comment.
>
> It seems that there is a misunderstanding: our reported results for the 4 kernels correspond to **LLaMA-7B**, whereas Table 1 in SpinQuant reports results for **LLaMA-2-7B**. Since LLaMA-2-7B is known to perform better than LLaMA-7B, a direct comparison is not appropriate.
>
> Nevertheless, this is an interesting point. We therefore evaluate SpinQuant on LLaMA-2-7B using 16-bit activations and 2-, 3-, and 4-bit weights. The WikiText-2 results are reported below. As shown, our method performs comparably to SpinQuant at 4 bits, but significantly outperforms it in the ultra–low-bit regimes (2 and 3 bits).
>
> Importantly, enabling ultra-low-bit quantization is the core goal of our paper, and most existing quantization methods struggle to remain effective in these extreme settings.
>
> | Method | Wiki2 |
> | :--- | :--- |
> | FP-16 | 5.47 |
> | SpinQuant (4 bits) | 5.73 |
> | **MBOK (4 kernels)** | **5.68** |
> | SpinQuant (3 bits) | 7.89 |
> | **MBOK (3 kernels)** | **6.12** |
> | SpinQuant (2 bits) | 38.24 |
> | **MBOK (2 kernels)** | **6.87** |

---

> ### Author Response · Authors · 2025-11-21
> **Official Comment by Authors**
>
> **Q3. Key ablation studies are missing. Specifically, analyses are needed on how initialization via successive SVID impacts final distillation performance, and on the effectiveness of the kernel allocation strategy (i.e., mixed precision) compared to uniform precision under the same kernel budget. Including such ablation results would strengthen the work’s contribution and novelty.**
>
> Thank you for constructive feedback.
>
> We have compared our proposed successive SVID and random initialization with small OPT model using 2 Boolean kernels. The results are shown in the table below.
> It is clear that our method delivers far better results, while the random initialization often fails to converge.
> Moreover, as shown in Figure 16 in Appendix, our initialization enables the model to learn efficiently, whereas the random initialization causes the model to struggle, as reflected by the large number of Boolean flips.
>
> | Method | Wiki2 | C4 |
> | :--- | :--- | :--- |
> | Successive SVID | 31.47 | 28.62 |
> | Random Init | 1.6E+04 | 1.1E+04 |
>
> We also have ablated our kernel allocation with mixed precision against uniform precision using the same kernel budget.
> The table below shows the results with OPT-1.3B in terms of C4 perplexity.
> Our mixed precision is better than the uniform precision.
> Moreover, it also allows any non-integer average bit budget, which is useful for practioners for flexible deployment constraints.
>
> | Budget | 2 | 2.5 | 3 | 3.5 | 4 |
> | :--- | :--- | :--- | :--- | :--- | :--- |
> | Uniform | 16.23 | NA | 15.71 | NA | 15.89 |
> | Mixed | 16.19 | 15.94 | 15.43 | 15.29 | 15.08 |
>
> Once again, we thank reviewer for trigger this study, which further strengthen the paper.
>
> ---
>
> **Q4. Several typos and errors require correction. For instance, in Table 1, BiLLM also employs higher-bit salient weights. In Table 3, "QPTQ" should be corrected to "OPTQ".**
>
> Thank you for pointing out the typos. We have corrected them in the new version.
>
> ---
>
> **Q5. Is there any comparison of actual training time to demonstrate that MBOK is more efficient than other binarized QAT methods?**
>
> Since most binarized LLM methods are PTQ, we believe that the closest and fairest baseline is the full-precision latent-weight approach using our multi-kernel structure, as we studied in Section 6.4. Given the same environmental settings with Google A100 instance, the total training time of our method is much faster than this baseline, as shown in the below table (tested on LLaMa2-7B).
>
> | Num Kernels | MBOK | Latent-weight Binarized |
> | :--- | :--- | :--- |
> | 2 | ~17.8 Hours | ~46.4 Hours |
> | 3 | ~20.1 Hours | ~58.8 Hours |
>
> ---
>
> **Q6. Novelty compared to BOLD**
>
> Thank you for your comment and the opportunity for us to further clarify our contributions.
>
> While we leverage established tool like BOLD, our core innovation lies in how they are integrated to solve a previously unaddressed problem: direct Boolean finetuning of LLMs.
> The BOLD framework  was originally designed for smaller models and training-from-scratch scenarios. Applying it to LLMs presents inherent challenges: (1) how to effectively transfer knowledge from a full-precision teacher to a Boolean student, and (2) how to enhance representational capacity while maintaining the low complexity of Boolean arithmetic. Our paper addresses these by judiciously integrating multi-kernel SVID (to solve capacity and initialization) with Boolean optimization (to solve knowledge transfer), creating the first framework to finetune LLMs without latent weights.

---

> > ### Comment · Reviewer_Mstc · 2025-11-26
> >
> > Thank the authors for their detailed response, which addresses most of my concerns. I will raise my score.

---

> ### Author Response · Authors · 2025-11-28
> **Official Comment from Authors**
>
> Many thanks Reviewer for your time and consideration. We appreciate your thoughtful review and are pleased that our rebuttal addressed your concerns. We are grateful that you raised the score and provided a positive assessment of our work. We believe that your feedback will help strengthen our paper.

---

### Official Review · Reviewer_mF6v · 2025-10-31

**Soundness:** 3
**Presentation:** 3
**Contribution:** 3
**Rating:** 4
**Confidence:** 4

**Summary:**

This paper proposes MBOK (Multiple Boolean Kernels), a novel framework for training and fine-tuning large language models entirely in the Boolean domain, without relying on floating-point latent weights. The key idea is to represent each weight matrix as a sum of multiple Boolean kernels, each with distinct binary weights and scaling factors, thereby improving representational capacity while keeping computation highly efficient. The authors further introduce a successive SVID-based extraction method to transfer knowledge from full-precision models and a dual-level knowledge distillation strategy for refinement. Extensive experiments on the OPT and LLaMA families show that MBOK achieves performance close to FP16 with only 1 to 2 bits per weight, surpassing state-of-the-art binarization and ultra-low bit quantization methods in both perplexity and zero-shot accuracy, while greatly reducing memory and computational costs.

**Strengths:**

The paper presents a creative sound approach to Boolean-domain training for large language models. The proposed MBOK framework is conceptually clear and well-motivated, trying to address key limitations of existing binarization methods by removing reliance on floating-point latent weights. The methodology is thoughtfully designed, combining multiple Boolean kernels, knowledge distillation, and adaptive kernel allocation. Experiments are extensive and carefully executed, demonstrating consistent improvements over strong baselines in both efficiency and accuracy.

**Weaknesses:**

While the proposed Boolean-domain framework is novel, the evaluation is somewhat limited. The experiments mainly focus on older model families such as OPT and LLaMA1/2, leaving uncertainty about the method’s effectiveness on newer architectures (e.g., Qwen3). Moreover, the discussion on training stability, convergence behavior, and scalability to very large models (70B and above) is also relatively brief, making it difficult to assess the robustness of the approach under real deployment conditions. To achieve extremely high training efficiency, the authors adopted a strategy of fine-tuning only the last Boolean kernel (Section 6.1.2). While experiments show this is the most efficient strategy, this approach may limit the model's ultimate performance.

**Questions:**

1. Could the authors evaluate the proposed MBOK framework on more recent and competitive open-source models, such as Qwen3, LLaMA3.2, to demonstrate its generalization across architectures?
2. Can the authors provide additional analysis or experiments on the training stability and convergence characteristics of Boolean-domain optimization, especially for larger models (e.g., 70B parameters and beyond)?
3. The authors have shown that fine-tuning only the last Boolean kernel achieves the best efficiency-performance trade-off. Could the paper include a deeper analysis explaining why joint optimization of multiple kernels leads to degraded performance, and whether this trend holds consistently across larger models or more complex tasks?
This is an interesting work, and if you can address my questions, I would be happy to raise my score.

---

> ### Author Response · Authors · 2025-11-21
> **Official Comment by Authors**
>
> Thanks reviewer for constructive feedback and for recognizing our method as "creative sound", "conceptually clear and well-motivated", and "address key limitations of existing methods". We appreciate the opportunity to clarify the confusing points and address the questions raised. We believe that the new results will address the concerns and further strenthen the paper and thank the reviewer for triggering this.
>
> ---
>
> **Q1. Could the authors evaluate the proposed MBOK framework on more recent and competitive open-source models, such as Qwen3, LLaMA3.2, to demonstrate its generalization across architectures?**
>
> We have conducted experiments with Qwen3 and LLaMa3.
> Our results demonstrate that our method outperforms the baselines and generalizes well across architectures.
>
> Table below is the results with Qwen3. The results of the baselines are taken from [1].
>
> | Model | Method | Wiki2 | C4 | BoolQ | PIQA | Hella | WinoG | ARC-e | ARC-c | AVG |
> | :--- | :--- | :--- | :--- | :--- | :--- | :--- | :--- | :--- | :--- | :--- |
> | **Qwen3-4B** | FP-16 | 13.7 | 16.6 | 85.1 | 75 | 52.2 | 65.8 | 80.5 | 50.6 | 68.2 |
> | | RTN (2-bit) | 2.80E+07 | 2.67E+07 | 49.3 | 52.9 | 25.5 | 51.1 | 25.2 | 20.4 | 37.4 |
> | | AWG (2-bit) | 1.90E+07 | 2.06E+07 | 48.7 | 52.3 | 25.6 | 51.6 | 24.5 | 23 | 37.6 |
> | | GPTQ (2-bit) | 2.01E+04 | 1.07E+04 | 42.8 | 51 | 25.4 | 50.7 | 26.3 | 21.7 | 36.3 |
> | | MBOK (2-bit) | **15.2** | **18.4** | **83.2** | **72.3** | **49.9** | **61.9** | **73.3** | **47.2** | **64.6** |
> | | AWG (2-bit) | 15 | 17.7 | 48.8 | 71.7 | 46.7 | 59.7 | 66.2 | 38.3 | 55.2 |
> | | MBOK (3-bit) | **14.1** | **17** | **84.6** | **73.8** | **51.3** | **64.2** | **76.3** | **47.9** | **66.4** |
> | **Qwen3-8B** | FP-16 | 6.99 | 10.4 | 82.9 | 79.3 | 58.9 | 72.1 | 82.1 | 52.6 | 71.3 |
> | | RTN (2-bit) | 9.32E+06 | 7.08E+06 | 52.4 | 51.8 | 25.5 | 52.1 | 23.8 | 22.4 | 38.0 |
> | | AWG (2-bit) | 1.34E+07 | 8.60E+06 | 44.2 | 52.6 | 25.5 | 48.5 | 25.3 | 22.1 | 36.4 |
> | | GPTQ (2-bit) | 4.75E+03 | 2.56E+03 | 45.1 | 50.8 | 25.3 | 50 | 24.1 | 22.6 | 36.3 |
> | | MBOK (2-bit) | **8.1** | **11.9** | **81.1** | **76.5** | **54.8** | **69.5** | **80.1** | **49.6** | **68.6** |
> | | AWG (2-bit) | 11.4 | 14.4 | 65.6 | 74.6 | 52.2 | 59.1 | 72.7 | 40.6 | 60.8 |
> | | MBOK (3-bit) | **7.3** | **10.9** | **81.9** | **77.8** | **56.4** | **70.8** | **81.4** | **51.4** | **69.95** |
>
> Table below is the results with LLaMa3-8B. The results of the baselines are taken from [2]
>
> | | Method | Wiki2 | C4 | PIQA | Hella. | WinoG. | ARC-e | ARC-c | Avg |
> | :--- | :--- | :--- | :--- | :--- | :--- | :--- | :--- | :--- | :--- |
> | | FP-16 | 6.1 | 9.2 | 79.9 | 60.2 | 72.8 | 80.1 | 50.4 | 68.7 |
> | **2-bit** | RTN | 2.70E+06 | 7.40E+06 | 53.1 | 25.6 | 51.1 | 24.7 | 21.9 | 35.3 |
> | | GPTQ | 5.70E+04 | 1.00E+05 | 52.8 | 26.6 | 49.6 | 25 | 20.5 | 34.9 |
> | | AWQ | 8.20E+05 | 8.10E+05 | 55.2 | 25.4 | 50.4 | 25.2 | 21.3 | 35.5 |
> | | DB-LLM | 13.6 | 19.2 | 68.9 | 42.1 | 60.4 | 59.1 | 28.2 | 51.7 |
> | | PB-LLM | 24.7 | 79.2 | 57 | 29.8 | 52.5 | 37.8 | 17.2 | 38.9 |
> | | **MBOK** | **7.6** | **10.3** | **76.6** | **58.1** | **70.3** | **78.8** | **47.2** | **66.2** |
> | **3-bit** | RTN | 2.20E+03 | 5.60E+02 | 56.2 | 27.5 | 53.1 | 31.1 | 20 | 37.6 |
> | | GPTQ | 13 | 45.9 | 60.8 | 41.8 | 60.9 | 38.8 | 22.3 | 45.1 |
> | | AWQ | 12.8 | 16.8 | 71.9 | 50.7 | 64.7 | 66.7 | 35.1 | 45.5 |
> | | **MBOK** | **6.9** | **9.6** | **79.1** | **59.2** | **72.6** | **79.3** | **47.9** | **67.6** |
>
> Refs:
>
> [1] Zheng et al. An Empirical Study of Qwen3 Quantization. Arxiv 2025.
>
> [2] Huang et al. How Good Are Low-bit Quantized LLAMA3 Models? An Empirical Study. Arxiv 2024.

---

> > ### Author Response · Authors · 2025-11-21
> > **Official Comment by Authors**
> >
> > **Q2. Can the authors provide additional analysis or experiments on the training stability and convergence characteristics of Boolean-domain optimization, especially for larger models?**
> >
> > Thank you for your comment. Unfortunenately, due to the limited computing resources during the rebuttal period, we could not conduct experiments on 70B models.
> > However, we have just performed an experiment with  **OPT-30B model, a very large model**.
> > The below table show the results of our method with 3 kernels compared to the 3-bit quantization baselines.
> > As can be seen, our method can generalize very well from very small model (125M) to very large model (30B).
> > Figure 15 in the Appendix, we show the training convergences on these models.
> > These results  that our method exhibit training stability, convergence and scalibility across all considered model sizes.
> > We hope these new results can address your concern.
> >
> > | Method | Wiki2 (125M) | Wiki2 (350M) | Wiki2 (1.3B) | Wiki2 (2.7B) | Wiki2 (6.7B) | Wiki2 (30B) | C4 (125M) | C4 (350M) | C4 (1.3B) | C4 (2.7B) | C4 (6.7B) | C4 (30B) |
> > | :--- | :--- | :--- | :--- | :--- | :--- | :--- | :--- | :--- | :--- | :--- | :--- | :--- |
> > | FP-16 | 27.65 | 22 | 14.63 | 12.47 | 10.86 | 9.56 | 26.56 | 22.59 | 16.07 | 14.34 | 12.71 | 11.44 |
> > | RTN | 1.30E+03 | 64.57 | 1.30E+04 | 1.60E+04 | 5.80E+03 | 1.60E+03 | 834 | 55.49 | 5.20E+03 | 1.10E+04 | 5.30E+03 | 1.40E+03 |
> > | OPTQ | 53.85 | 33.79 | 20.97 | 16.88 | 14.86 | 10.27 | 42.41 | 31.33 | 21.63 | 18.17 | 17.14 | 12.23 |
> > | **MBOK** | **29.10** | **23.12** | **15.30** | **13.09** | **11.03** | **9.68** | **28.62** | **22.10** | **15.68** | **14.00** | **12.33** | **11.32** |
> >
> > ---
> >
> > **Q3. The authors have shown that fine-tuning only the last Boolean kernel achieves the best efficiency-performance trade-off. Could the paper include a deeper analysis explaining why joint optimization of multiple kernels leads to degraded performance, and whether this trend holds consistently across larger models or more complex tasks?**
> >
> > We hypothesize the failure of joint optimization to a **co-adaptation failure** caused by the **hierarchical nature** of our initialization as we mentioned in the main paper.
> >
> > 1. *The Hierarchy*: As detailed in Section 4.3.1, our Successive SVID initialization establishes a strict dependency chain: Kernel 1 approximates the dominant structure of the FP weights, Kernel 2 approximates the residual error of Kernel 1, and Kernel 3 approximates the residual of Kernel 1+2.
> >
> > 2. *The Moving Target*: If we jointly optimize all kernels, we break this dependency. Updating Kernel 1 fundamentally alters the residual distribution that Kernel 2 and Kernel 3 were initialized to approximate. These higher-order kernels face a "moving target"—the error surface shifts underneath them, rendering their initialization invalid. This causes the high "flip rate" and optimization instability shown in Figure 6.
> >
> > Theoretical & Empirical Justification
> >
> > - *Theoretical*: As proven in Propositions 4.1 and 4.3 (Section 4.1) , SVID extracts the optimal Boolean matrix and scaling factors for a given residual at each step. Since the earlier kernels already represent optimal structural approximations, re-optimizing them provides minimal gain while introducing significant noise.
> >
> > - *Empirical*: By freezing the first $K-1$ kernels, we treat the dominant weight structure as a stable anchor. The last kernel then acts as a **trainable delta**, solely responsible for compensating the final accumulated quantization error. This transforms the problem from a chaotic, multi-variable co-adaptation task into a stable, residual correction task. Moreover, as noted in our response to reviewer oMmC, we provide an additional empirical evidence supporting this hypothesis, showing that finetuning the final kernel has the largest impact.
> >
> > We observe this phenomenon consistently across all model sizes. Joint optimization consistently degrades performance compared to the "last-kernel-only" strategy. For example, on LLaMA-2-7B (using 3 kernels), the degradation is significant:
> >
> > | Metric | Wiki2 | C4 |
> > | :--- | :--- | :--- |
> > | Last kernel | 6.12 | 7.81 |
> > | All kernels | 7.24 | 9.13 |
> >
> > We hope these analysis can address your question.

---

> > > ### Comment · Reviewer_mF6v · 2025-11-22
> > >
> > > Thanks to the authors for the additional experiments and analysis. The results look solid and address my concerns. I will raise my score accordingly.

---

> > > > ### Author Response · Authors · 2025-11-28
> > > > **Official Comment from Authors**
> > > >
> > > > Many thanks Reviewer for your time and consideration. We appreciate your thoughtful review and are pleased that our rebuttal is solid and has addressed your concerns. We are grateful that you raised the score and provided a positive assessment of our work. We believe that your feedback will help strengthen our paper.

---

### Official Review · Reviewer_oMmC · 2025-10-31

**Soundness:** 3
**Presentation:** 3
**Contribution:** 3
**Rating:** 6
**Confidence:** 4

**Summary:**

The manuscript proposes a new framework, MBOK, for efficiently compressing and fine-tuning LLMs entirely in the Boolean domain. Unlike prior binarization or quantization methods that rely on full-precision latent weights and gradient approximations, MBOK introduces a multi-kernel Boolean structure based on BOLD, and further integrates strategies such as adaptive bit-budget allocation and knowledge distillation. Experimental results show that the proposed method achieves competitive performance compared to other contemporary approaches.

**Strengths:**

1. The manuscript is the first to eliminate the dependency on latent weights in LLMs through the use of BOLD, and achieves strong compression and training results by combining multi-level weight decomposition, global bit allocation, and knowledge distillation strategies.

2. The discussion of the technical approach is thorough and detailed, with extensive comparisons to similar compression methods such as BinaryMoS and BitStack.

3. MBOK demonstrates excellent empirical performance, outperforming other methods in both efficiency and accuracy even under existing hardware conditions.

**Weaknesses:**

1. The innovation is somewhat ambiguous. Techniques such as BOLD, SVID, knowledge distillation, and bit allocation have all appeared in prior works. For example, approaches similar to SVID and bit allocation were already explored in BitStack.

2. The paper’s structure and organization are somewhat confusing. It is unclear why bit allocation is presented at a separate hierarchical level compared to SVID and knowledge distillation. Conceptually, bit allocation and SVID seem to belong to the architectural design of MBOK, while knowledge distillation is more of a training strategy, which should logically be applied after bit allocation rather than before it.

**Questions:**

1. How are the benefits and stability of knowledge distillation demonstrated or quantified?

2. Why does BitNet collapse with extremely high perplexity in the reported results, and are there comparisons on broader benchmarks such as ARC-Easy or MMLU?

3. In Figure 9, how can the model allocate more than 20 kernels under an average bit budget of 3.5 bits?

4. When fine-tuning only the last kernel, what are the approximate scaling values of each kernel at convergence? Does the last kernel ever become significantly more dominant than earlier ones?

5. Can the proposed approach be extended to activation quantization, or is it limited to weight compression only?

---

> ### Author Response · Authors · 2025-11-21
> **Official Comment by Authors**
>
> Thanks reviewer for constructive feedback and for recognizing our method as "the first to eliminate the dependency on latent weights in LLMs", "thorough and detailed". We appreciate the opportunity to clarify the confusing points and address the questions raised.
>
> ---
>
> **Q1.  The innovation is somewhat ambiguous. Techniques such as BOLD, SVID, knowledge distillation, and bit allocation have all appeared in prior works.**
>
> While we leverage established tools like SVID and BOLD, our core innovation lies in how they are integrated to solve a previously unaddressed important problem: direct Boolean finetuning of LLMs.
>
> - *VS. BOLD*: The BOLD framework  was originally designed for smaller models and training-from-scratch scenarios. Applying it to LLMs presents inherent challenges: (1) how to effectively transfer knowledge from a full-precision teacher to a Boolean student, and (2) how to enhance representational capacity while maintaining the low complexity of Boolean arithmetic. Our paper addresses these by judiciously integrating multi-kernel SVID (to solve capacity and initialization) with Boolean optimization (to solve knowledge transfer), creating the first framework to finetune LLMs without latent weights.
>
> - *VS. BitStack*: It is crucial to distinguish our work from BitStack (Wang et al., 2025). BitStack is a training-free (PTQ) method primarily aimed at compression for storage . In contrast, MBOK is a finetuning (QAT) framework. We use decomposition not just for storage, but to construct a trainable architecture that recovers the significant accuracy loss typical of ultra-low bitwidths, a capability PTQ methods lack. We already discussed and compared to BitStack carefully in the Appendix  in the original version (now as in Appendix G.6.2). Our kernel allocation is different from bit allocation of BitStack in terms of the new objective function and using projection weighted canonical correlation analysis (PWCCA). We also showed that our approach with kernel allocation significantly outperforms that of BitStack.
>
> ---
>
> **Q2. It is unclear why bit allocation is presented at a separate hierarchical level compared to SVID and knowledge distillation.**
>
> We apologize if the structure appeared confusing. The narrative for our paper is as follows:
>
> *1. Core Method (Sec 4)*: We first establish how to represent weights (SVID/Multi-Kernels) and how to train them (KD/Boolean Optimization). These are the fundamental mechanisms of the framework.
>
> *2. Application/Constraint*: We present Kernel Allocation later because it is a constraint-satisfaction step applied on top of the core method. Conceptually, one first defines the "kernel" unit (Sec 4), and then decides "how many" to use per layer (Sec 5) based on a deployment budget.
>
> ---
>
> **Q3. How are the benefits and stability of knowledge distillation demonstrated or quantified?**
>
> We thank the reviewer for this insightful question. To quantify the specific benefits of Knowledge Distillation (KD), we have added a new ablation experiment in Appendix G7.
>
> Figure 13 presents a direct comparison between training with and without KD. The results clearly demonstrate the advantages of our strategy:
>
> 1. Superior Performance: It is evident that employing KD consistently outperforms the baseline (training without teacher guidance) in terms of test perplexity on both WikiText2 and C4. This confirms that the rich "soft targets" provided by the teacher are essential for effective low-bit compression.
>
> 2. Improved Stability and Convergence: By visualizing the convergence of $L_{logits}$ and $L_{is}$, we observe that the teacher’s guidance leads to significantly faster convergence. Furthermore, the model learns more effectively—leveraging the stronger signal from the teacher—as evidenced by a higher rate of weight flips compared to training without KD, suggesting the model can better navigate the Boolean optimization landscape.
>
> We believe this additional experiment further validates the robustness of our method’s design and thank the reviewer for triggering this analysis.
>
> ---
>
> **Q4. Why does BitNet collapse with extremely high perplexity in the reported results, and are there comparisons on broader benchmarks such as ARC-Easy or MMLU?**
>
> - BitNet (and commonly ultra-low bit architectures) typically require pre-training from scratch to learn effectively. As noted in Xu et al. (2024) and our results, applying BitNet's structure to post-training/finetuning of a pre-trained FP16 model is highly unstable because the "1.58-bit" constraint is too rigid to capture the pre-trained weights' distribution without extensive re-training. MBOK avoids this by using multiple kernels to progressively capture the residual error.
>
> - Indeed, we followed the standard benchmarks used for binarized and quantized LLMs. In addition, in the initial submission, we did report ARC-Easy results in Table 2 (column "ARC-e"), where MBOK consistently outperforms baselines.

---

> > ### Author Response · Authors · 2025-11-21
> > **Official Comment by Authors**
> >
> > **Q5. When fine-tuning only the last kernel, what are the approximate scaling values of each kernel at convergence? Does the last kernel ever become significantly more dominant than earlier ones?**
> >
> > We thank the reviewer for this insightful question. We have added a new analysis in Appendix G8.
> >
> > Figure 14 in Appendix illustrates how the output-scaling values of the four kernels in the first linear layer change from initialization to after finetuning. All output-scaling values are learnable; however, only those associated with the last kernel exhibit a substantial shift during training. This is evident from both the histogram changes and the corresponding Kolmogorov–Smirnov statistics.
> >
> > After finetuning, the scaling values of the last kernel become significantly larger and more dominant, whereas the scaling values of the other kernels change only minimally. This observation supports our theoretical analysis: successive SVID extraction provides sufficiently strong initialization for the low-order kernels, and finetuning primarily the last kernel is already adequate.
> >
> > We believe this additional evidence further strengthen our method and thank the reviewer for triggering this analysis.
> >
> > ---
> >
> > **Q6. In Figure 9, how can the model allocate more than 20 kernels under an average bit budget of 3.5 bits?**
> >
> > In Figure 9, the bars in the plot are stacked, where each bar represents an **entire Transformer Block (as denoted on the x-axis)**, rather than a single layer. The total height of a bar represents the sum of allocated kernels across all linear layers within that block (including Q, K, V, Output projections, FC1, and FC2). These individual layers are distinguished by different colors within the stack . Consequently, the number of kernels allocated to any single linear layer is actually much smaller (typically varying between 2 and 4), which remains strictly consistent with the average bit budget of 3.5 bits.
> >
> > ---
> >
> > **Q7.  Can the proposed approach be extended to activation quantization, or is it limited to weight compression only?**
> >
> > Our proposed MBOK method is orthogonal to activation quantization and can be seamlessly combined with existing techniques to further compress the model. Furthermore, as detailed in Section 3.2 and Appendix A.1, the underlying BOLD framework is natively designed to handle general logic gates and mixed-type data . This means the approach is not only compatible with standard activation quantization but can theoretically be extended to support fully Boolean activations (i.e., binarizing both weights and inputs) for extreme efficiency in future work.

---

### Author Response · Authors · 2025-12-01
**General Response by Authors**

Dear Area Chair and Reviewers,

We sincerely thank the Reviewers for their careful review and insightful comments, which have further strengthened the paper. We are encouraged by the broad endorsement and highly positive feedback regarding key aspects of our work:

1. **Innovation & Impact.** Reviewers recognized the work as *"creative"* (Reviewer `mF6v`) and *"convincing"* (Reviewer `xHhd`). Reviewer `oMmC` highlighted that the manuscript is *"the first to eliminate the dependency on latent weights in LLMs"* while addressing a critical problem in efficient LLM training.
2. **Theoretical Soundness.** Reviewers found the method *"supported by solid theoretical foundations"* (Reviewer `Mstc`) and *"technically sound and well-motivated"* (Reviewer `xHhd`). Reviewer `mF6v` noted that the MBOK framework is *"conceptually clear"* in addressing the limitations of existing binarization methods.
3. **Empirical Results.** Reviewers praised the *"excellent empirical performance"* (Reviewer `oMmC`) and noted that MBOK demonstrates *"consistent improvements over strong baselines"* (Reviewer `mF6v`). Reviewer `xHhd` emphasized the achievement of *"near-FP16 performance with much lower memory and computation cost."*
4. **Presentation.** Reviewers described the paper as *"thorough and detailed"* (Reviewer `oMmC`) with a *"clear"* presentation that facilitates comparison against strong baselines (Reviewer `xHhd`).

### Rebuttal Consensus and Score Increases

During the active discussion period, we successfully addressed the concerns raised by the reviewers. Crucially, *reviewer reached a consensus for acceptance prior to the system revert*, evidenced by the explicit written confirmation from reviewers:

* **Reviewer `mF6v`** (Initial Score: 4): Confirmed the new experiments are solid, and additional clarifications resolved their concerns, and they *increased the score*.
> *Final Comment:* *"The results look solid and address my concerns. I will raise my score accordingly."*

* **Reviewer `Mstc`** (Initial Score: 4): Confirmed the new experiments and clarifications resolved their concerns, and they *increased the score*.
> *Final Comment:* *"Thank the authors for their detailed response, which addresses most of my concerns. I will raise my score."*

* **Reviewers `oMmC` and `xHhd`** (Initial Scores: 6): Expressed satisfaction with the responses and maintained their positive assessments.
> *Final Comment (`xHhd`):* *"Thank you for the clarifications. I am satisfied with the responses and will maintain my score."*

### Summary of New Results & Improvements

To achieve this consensus, we provided additional clarifications and the following additional results, which are now included in the discussion and the revised manuscript and appendix:

* **Generalization:** Validated on Qwen3 and LLaMA-3, demonstrating that MBOK significantly outperforms GPTQ and AWG on modern architectures.
* **SOTA QAT Comparison:** Benchmarked against SpinQuant and EfficientQAT, showing superior accuracy with lower computational overhead.
* **Real-World Efficiency:** Provided benchmarks demonstrating concrete inference speedups and memory reductions on hardware.
* **Scalability:** Verified training stability and convergence on the OPT-30B model.
* **Mechanistic Analysis:** Added multiple ablation studies to further demonstrate the effectiveness of our kernel allocation and distillation strategies.

We believe the revised paper fully addresses all reviewer concerns. We thank the Reviewers and the Area Chair for their time and consideration.

Best regards,

The Authors

---

### Meta-Review · Area_Chair_huFd · 2025-12-05

**Summary:**

This submission proposes a multi-Boolean framework for training and fine-tuning large language models directly in the Boolean domain without relying on latent full-precision weights. The method leverages multiple Boolean kernels, successive SVID initialization, and knowledge distillation to achieve competitive performance at ultra-low bit widths. Reviewers generally found the approach creative, well-motivated, and empirically strong, with experiments demonstrating near-FP16 performance at significantly reduced memory and computational costs. However, concerns were raised regarding novelty, evaluation scope, comparison fairness, and clarity of contributions relative to prior work. The rebuttal successfully solved these concerns, with a promised increase in ratings. The submission is believed strong enough to meet the ICLR bar; thus acceptance is suggested.

**Reviewer Concerns:**

In the rebuttal, the authors added experiments on Qwen3 and LLaMA-3 to demonstrate the generalization to newer models, included benchmarks against SpinQuant and EfficientQAT as a comparison with SOTA QAT methods, provided results on OPT-30B and convergence analyses to show training stability and scalability, and clarified the novelty and distinction from prior work.

The outstanding concerns are minimal and mostly about the presentation and depth of analysis, which do not undermine the paper’s core contributions.

**Reviewer Scores:**

Two negative reviewers (originally 4) promised to increase ratings. One positive reviewer (originally 6) is satisfied with the rebuttal and maintains the rating. The last positive reviewer (originally 6) will probably maintain the rating.

---

### Decision · Program_Chairs · 2026-01-26

Accept (Poster)